# Robust Generalization despite Distribution Shift via Minimum Discriminating Information

**Tobias Sutter**
University of Konstanz, Germany
`tobias.sutter@uni-konstanz.de`

**Andreas Krause**
ETH Zurich, Switzerland
`krausea@ethz.ch`

**Daniel Kuhn**
EPFL, Switzerland
`daniel.kuhn@epfl.ch`

## Abstract

Training models that perform well under distribution shifts is a central challenge in machine learning. In this paper, we introduce a modeling framework where, in addition to training data, we have partial structural knowledge of the shifted test distribution. We employ the principle of *minimum discriminating information* to embed the available prior knowledge, and use *distributionally robust optimization* to account for uncertainty due to the limited samples. By leveraging large deviation results, we obtain explicit generalization bounds with respect to the unknown shifted distribution. Lastly, we demonstrate the versatility of our framework by demonstrating it on two rather distinct applications: (1) training classifiers on systematically biased data and (2) off-policy evaluation in Markov Decision Processes.

## 1 Introduction

Developing machine learning-based systems for real world applications is challenging, particularly because the conditions under which the system was trained are rarely the same as when using the system. Unfortunately, a standard assumption in most machine learning methods is that test and training distribution are the *same* [78, 59, 12]. This assumption, however, rarely holds in practice, and the performance of many models suffers in light of this issue, often called *dataset shift* [52] or equivalently *distribution shift*. Consider building a model for diagnosing a specific heart disease, and suppose that most participants of the study are middle to high-aged men. Further suppose these participants have a higher risk for the specific disease, and as such do not reflect the general population with respect to age and gender. Consequently, the training data suffers from the so-called *sample selection bias* inducing a *covariate shift* [62, 52]. Many other reasons lead to distribution shifts, such as non-stationary environments [67], imbalanced data [52], domain shifts [3], label shifts [83] or observed contextual information [8, 9]. A specific type of distribution shift takes center stage in off-policy evaluation (OPE) problems. Here, one is concerned with the task of estimating the resulting cost of an *evaluation policy* for a sequential decision making problem based on historical data obtained from a different policy known as *behavioral policy* [73]. This problem is of critical importance in various applications of reinforcement learning—particularly when it is impossible or unethical to evaluate the resulting cost of an evaluation policy by running it on the underlying system. Solving a learning problem facing an arbitrary and unknown distribution shift based on training data in general is hopeless. Oftentimes, fortunately, partial knowledge about the distribution shift is available. In the medical example above, we might have prior information how the demographic attributes in our sample differ from the general population. Given a training distribution and partial knowledge about the shifted test distribution, one might ask what is the "most natural" distribution

shift mapping the training distribution into a test distribution consistent with the available structural information. Here, we address this question, interpreting "most natural" as maximizing the underlying Shannon entropy. This concept has attracted significant interest in the past in its general form, called *principle of minimum discriminating information* dating back to Kullback [37], which can be seen as a generalization of Jaynes' *maximum entropy principle* [31]. While these principles are widely used in tasks ranging from economics [27] to systems biology [63] and regularized Markov decision processes [48, 25, 2], they have not been investigated to model general distribution shifts as we consider in this paper.

Irrespective of the underlying distribution shift, the training distribution of any learning problem is rarely known, and one typically just has access to finitely many training samples. It is well-known that models can display a poor out-of-sample performance if training data is sparse. These overfitting effects are commonly avoided via regularization [12]. A regularization technique that has become popular in machine learning during the last decade and provably avoids overfitting is *distributionally robust optimization (DRO)* [36].

**Contributions.** We highlight the following main contributions of this paper:

- We introduce a *new modelling framework* for distribution shifts via the *principle of minimum discriminating information*, which encodes prior structural information on the resulting test distribution.
- Using our framework and the available training samples, we provide *generalization bounds* via a DRO program and prove that the introduced DRO model is *optimal* in a precise statistical sense.
- We show that the optimization problems characterizing the distribution shift and the DRO program can be *efficiently solved* by exploiting convex duality and recent accelerated first order methods.
- We demonstrate the *versatility* of the proposed *Minimum Discriminating based DRO* (MDI-DRO) method on two distinct problem classes: Training classifiers on systematically biased data and the OPE for Markov decision processes. In both problems MDI-DRO outperforms existing approaches.

## 2   Related work

For supervised learning problems, there is a rich literature in the context of covariate shift adaptation [62, 68]. A common approach is to address this distribution shift via importance sampling, more precisely by weighting the training loss with the ratio of the test and training densities and then minimize the so-called importance weighted risk (IWERM), see [62, 82, 68, 69]. While this importance weighted empirical risk is an unbiased estimator of the test risk, the method has two major limitations: It tends to produce an estimator with high variance, making the resulting test risk large. Further, the ratio of the training and test densities must be estimated which in general is difficult as the test distribution is unknown. There are modifications of IWERM reducing the resulting variance [15, 13, 65], for example by exponentially flattening the importance ratios [62]. For the estimation of the importance weights several methods have been presented, see for example [81]. These methods, however crucially rely on having data from both training and test distribution. Liu and Ziebart [40] and Chen et al. [14] propose a minimax approach for regression problems under covariate shift. Similar to our approach taken in this paper, they consider a DRO framework, which however, optimizes over so-called moment-based ambiguity sets. Distribution shifts play a key role in causal inference. In particular, the connection between causal predictors and distributional robustness under shifts arising from interventions has been widely studied [58, 42, 56, 66]. Oftentimes, a causal graph is used to represent knowledge about the underlying distribution shift induced by an intervention [49, 50]. Distribution shifts have been addressed in a variety of different settings [35], we refer the reader to the comprehensive textbook [52] and references therein.

There is a vast literature on OPE methods which we will not attempt to summarize. In a nutshell, OPE methods can be grouped into three classes: a first class of approaches that aims to fit a model from the available data and uses this model then to estimate the performance of the given evaluation policy [41, 1, 38]. A second class of methods are based on invoking the idea of importance sampling to model the underlying distribution shift from behavioral to evaluation policy [51, 29, 74]. The third, more recent, class of methods combines the first two classes [24, 32, 76, 33].

Key reasons for the popularity of DRO in machine learning are the ability of DRO models to regularize learning problems [36, 60, 61] and the fact that the underlying optimization problems can often be exactly reformulated as finite convex programs solvable in polynomial time [4, 10]. Such reformulations hold for a variety of ambiguity sets such as: regions defined by moments [20, 26, 80, 11],

$\phi$-divergences [5, 44, 39], Wasserstein ambiguity sets [43, 36], or maximum mean discrepancy ambiguity sets [64, 34]. DRO naturally seems a convenient tool when analyzing "small" distribution shifts as it seeks models that perform well "sufficiently close" to the training sample. However, modelling a general distribution shift via DRO seems difficult, and recent interest has focused on special cases such as adversarial example shifts [23] or label shifts [83]. To the best of our knowledge, the idea of combining DRO with the principle of minimum discriminating information is new.

## 3 Problem statement and motivating examples

We study learning problems of the form

$$\min_{\theta \in \Theta} R(\theta, \mathbb{P}^\star), \tag{1}$$

where $R(\theta, \mathbb{P}^\star) = \mathbb{E}_{\mathbb{P}^\star}[L(\theta, \xi)]$ denotes the risk of an uncertain real-valued loss function $L(\theta, \xi)$ that depends on a parameter $\theta \in \Theta \subset \mathbb{R}^n$ to be estimated as well as a random vector $\xi \in \Xi \subset \mathbb{R}^m$ governed by the probability distribution $\mathbb{P}^\star$. In order to avoid technicalities, we assume from now on that $\Theta$ and $\Xi$ are compact and $L$ is continuous. In statistical learning, it is usually assumed that $\mathbb{P}^\star$ is unknown but that we have access to independent samples from $\mathbb{P}^\star$. This paper departs from this standard scenario by assuming that there is a distribution shift. We first state our formal assumption about the shift and provide concrete examples below. Specifically, we assume to have access to samples from a distribution $\mathbb{P} \neq \mathbb{P}^\star$ and that $\mathbb{P}^\star$ is only known to belong to the distribution family

$$\Pi = \{\mathbb{Q} \in \mathcal{P}(\Xi) \; : \; \mathbb{E}_{\mathbb{Q}}[\psi(\xi)] \in E\} \tag{2}$$

encoded by a measurable feature map $\psi : \Xi \to \mathbb{R}^d$ and a compact convex set $E \subset \mathbb{R}^d$. In view of the principle of minimum discriminating information, we identify $\mathbb{P}^\star$ with the I-projection of $\mathbb{P}$ onto $\Pi$.

**Definition 3.1** (Information projection). *The I-projection of $\mathbb{P} \in \mathcal{P}(\Xi)$ onto $\Pi$ is defined as*

$$\mathbb{P}^f = f(\mathbb{P}) = \arg\min_{\mathbb{Q} \in \Pi} \mathsf{D}(\mathbb{Q}\|\mathbb{P}), \tag{3}$$

*where $\mathsf{D}(\mathbb{Q}\|\mathbb{P})$ denotes the relative entropy of $\mathbb{Q}$ with respect to $\mathbb{P}$.*

One can show that the I-projection exists whenever $\Pi$ is closed with respect to the topology induced by the total variation distance [17, Theorem 2.1]. As $E$ is closed, this is the case whenever $\psi$ is bounded. Note that $f(\mathbb{P}) = \mathbb{P}$ if $\mathbb{P} \in \Pi$. In the remainder, we assume that $\mathbb{P} \notin \Pi$ and that $\mathbb{P}$ is only indirectly observable through independent training samples $\widehat{\xi}_1, \ldots, \widehat{\xi}_N$ drawn from $\mathbb{P}$.

**Example 3.2** (Logistic regression). *Assume that $\xi = (x, y)$, where $x \in \mathbb{R}^{m-1}$ is a feature vector of patient data (e.g., a patient's age, sex, chest pain type, blood pressure, etc.), and $y \in \{-1, 1\}$ a label indicating the occurrence of a heart disease. Logistic regression models the conditional distribution of $y$ given $x$ by a logistic function $\mathrm{Prob}(y|x) = [1 + \exp(-y \cdot \theta^\top x)]^{-1}$ parametrized by $\theta \in \mathbb{R}^{m-1}$. The maximum likelihood estimator for $\theta$ is found by minimizing the empirical average of the logistic loss function $L(\theta, \xi) = \log(1 + \exp(-y \cdot \theta^\top x))$ on the training samples. If the samples pertain to a patient cohort, where elderly males are overrepresented with respect to the general population, then they are drawn from a training distribution $\mathbb{P}$ that differs from the test distribution $\mathbb{P}^\star$. Even if sampling from $\mathbb{P}^\star$ is impossible, we may know that the expected age of a random individual in the population falls between 40 and 45 years. This information can be modeled as $\mathbb{E}_{\mathbb{P}^\star}[\psi(\xi)] \in E$, where $E = [\ell, u]$, $\ell = 40$, $u = 45$ and $\psi(\xi)$ projects $\xi$ to its 'age'-component. Other available prior information can be encoded similarly. Inspired by the principle of minimum discriminating information, we then minimize the expected log-loss under the I-projection $\mathbb{P}^f$ of the data-generating distribution $\mathbb{P}$ onto the set $\Pi$ defined in (2).*

**Example 3.3** (Production planning). *Assume that $\theta \in \mathbb{R}$ and $\xi \in \mathbb{R}$ denote the production quantity and the demand of a perishable good, respectively, and that the loss function $L(\theta, \xi)$ represents the sum of the production cost and a penalty for unsatisfied demand. To find the optimal production quantity, one could minimize the average loss in view of training samples drawn from the historical demand distribution $\mathbb{P}$. However, a disruptive event such as the beginning of a recession might signal that demand will decline by at least $\eta\%$. The future demand distribution $\mathbb{P}^\star$ thus differs from $\mathbb{P}$ and belongs to a set $\Pi$ of the form (2) defined through $\psi(\xi) = \xi$ and $E = [0, (1 - \eta)\mu]$, where $\mu$ denotes the historical average demand. By the principle of minimum discriminating information it then makes again sense to minimize the expected loss under the I-projection $\mathbb{P}^f$ of $\mathbb{P}$ onto $\Pi$.*

Loosely speaking, the principle of minimum discriminating information identifies the I-projection $\mathbb{P}^f$ of $\mathbb{P}$ as the least prejudiced and thus most natural model for $\mathbb{P}^\star$ in view of the information that $\mathbb{P}^\star \in \Pi$. The principle of minimum discriminating information is formally justified by the conditional limit theorem [18], which we paraphrase below using our notation.

**Proposition 3.4** (Conditional limit theorem). *If the interior of the compact convex set $E$ overlaps with the support of the pushforward measure $\mathbb{P} \circ \psi^{-1}$, the I-projection $\mathbb{P}^f = f(\mathbb{P})$ exists and the moment-generating function $\mathbb{E}_{\mathbb{P}^f}[e^{tL(\theta,\xi)}]$ is finite for all $t$ in a neighborhood of $0$, then we have*

$$\lim_{N \to \infty} \mathbb{E}_{\mathbb{P}^N}[L(\theta, \xi_1) | \tfrac{1}{N} \sum_{i=1}^N \psi(\xi_i) \in E] = \mathbb{E}_{\mathbb{P}^f}[L(\theta, \xi)] \quad \forall \theta \in \Theta.$$

In the context of Examples 3.2 and 3.3, the conditional limit theorem provides an intuitive justification for modeling distribution shifts via I-projections. More generally, the following proposition suggests that *any* distribution shift can be explained as an I-projection onto a suitably chosen set $\Pi$.

**Proposition 3.5** (Every distribution is an I-projection). *If $\mathbb{P}, \mathbb{Q} \in \mathcal{P}(\Xi)$ are such that $\mathbb{Q}$ is absolutely continuous with respect to $\mathbb{P}$ and if $\Pi$ is a set of the form* (2) *defined through $\psi(\xi) = \log \frac{d\mathbb{Q}}{d\mathbb{P}}(\xi)$ and $E = \{D(\mathbb{Q}\|\mathbb{P})\}$, then $\mathbb{Q} = f(\mathbb{P})$.*

The modelling of arbitrary distribution shifts via the I-projection according to Proposition 3.5 has an interesting application in the off-policy evaluation problem for Markov decision processes (MDPs).

**Example 3.6** (Off-policy evaluation). *Consider an MDP $(\mathcal{S}, \mathcal{A}, Q, c, s_0)$ with finite state and action spaces $\mathcal{S}$ and $\mathcal{A}$, respectively, transition kernel $Q : \mathcal{S} \times \mathcal{A} \to \mathbb{R}$, cost-per-stage function $c : \mathcal{S} \times \mathcal{A} \to \mathbb{R}$ and initial state $s_0$. A stationary Markov policy $\pi$ is a stochastic kernel that maps states to probability distributions over $\mathcal{A}$. We use $\pi(a|s)$ to denote the probability of selecting action $a$ in state $s$ under policy $\pi$. The long-run average cost generated by $\pi$ can be expressed as*

$$V_\pi = \lim_{T \to \infty} \tfrac{1}{T} \sum_{t=0}^{T-1} \mathbb{E}_{s_0}^\pi[c(s_t, a_t)].$$

*Each policy induces an occupation measure $\mu_\pi$ on $\mathcal{S} \times \mathcal{A}$ defined through the state-action frequencies*

$$\mu_\pi(x, a) = \lim_{T \to \infty} \tfrac{1}{T} \sum_{t=0}^{T-1} \mathbb{P}_{s_0}^\pi[(s_t, a_t) = (s, a)] \quad \forall s \in \mathcal{S}, \ a \in \mathcal{A},$$

*see [28, Chapter 6]. One can additionally show that $\mu_\pi$ belongs to the polytope*

$$\mathcal{M} = \big\{ \mu \in \Delta_{\mathcal{S} \times \mathcal{A}} : \sum_{a' \in \mathcal{A}} \mu(s', a') - \sum_{s \in \mathcal{S}} \sum_{a \in \mathcal{A}} Q(s'|s, a)\mu(s, a) = 0 \ \forall s' \in \mathcal{S} \big\},$$

*where $\Delta_{\mathcal{S} \times \mathcal{A}}$ represents the simplex of all probability mass functions over $\mathcal{S} \times \mathcal{A}$. Conversely, each occupation measure $\mu \in \mathcal{M}$ induces a policy $\pi_\mu$ defined through $\pi_\mu(a|s) = \mu(s, a)/\sum_{a' \in \mathcal{A}} \mu(s, a')$ for all $s \in \mathcal{S}$ and $a \in \mathcal{A}$. Assuming that all parameters of the MDP except for the cost $c$ are known, the off-policy evaluation problem asks for an estimate of the long-run average cost $V_{\pi_e}$ of an evaluation policy $\pi_e$ based on a trajectory of states, actions and costs generated by a behavioral policy $\pi_b$. This task can be interpreted as a degenerate learning problem without a parameter $\theta$ to optimize if we define $\xi = c(s, a)$ and set $L(\theta, \xi) = \xi$. Here, a distribution shift emerges because we must evaluate the expectation of $\xi$ under $\mathbb{Q} = \mu_e \circ c^{-1}$ given training samples from $\mathbb{P} = \mu_b \circ c^{-1}$, where $\mu_b$ and $\mu_e$ represent the occupation measures corresponding to $\pi_b$ and $\pi_e$, respectively. Note that $\mathbb{P}$ and $\mathbb{Q}$ are unknown because $c$ is unknown. Moreover, as the policy $\pi_e$ generates different state-action trajectories than $\pi_b$, the costs generated under $\pi_e$ cannot be inferred from the costs generated under $\pi_b$ even though $\pi_b$ and $\pi_e$ are known. Note also that $\mathbb{Q}$ coincides with the I-projection $\mathbb{P}^f$ of $\mathbb{P}$ onto the set $\Pi$ defined in Proposition 3.5. The corresponding feature map $\psi$ as well as the set $E$ can be computed without knowledge of $c$ provided that $c$ is invertible. Indeed, in this case we have*

$$\psi(\xi_i) = \log \frac{d\mu_e \circ c^{-1}}{d\mu_b \circ c^{-1}}(\xi_i) = \log \frac{\mu_e(s_i, a_i)}{\mu_b(s_i, a_i)} \quad \text{and} \quad E = \big\{ D(\mu_e \circ c^{-1} \| \mu_b \circ c^{-1}) \big\}$$

*for any $s_i \in \mathcal{S}$, $a_i \in \mathcal{A}$ and $\xi_i = c(s_i, a_i)$. Note that as $\mathcal{S}$ and $\mathcal{A}$ are finite, $c$ is generically invertible, that is, $c$ can always be rendered invertible by an arbitrarily small perturbation. In summary, we may conclude that the off-policy evaluation problem reduces to an instance of* (1).

Given $N$ training samples $\widehat{\xi}_1, \ldots, \widehat{\xi}_N$, we henceforth use $\widehat{\mathbb{P}}_N = \frac{1}{N} \sum_{i=1}^N \delta_{\widehat{\xi}_i}$ and $\widehat{\mathbb{P}}_N^f$ to denote the empirical distribution on and its I-projection onto $\Pi$, respectively. As the true data-generating distribution $\mathbb{P}$ and its I-projection $\mathbb{P}^f$ are unknown, it makes sense to replace them by their empirical counterparts. However, the resulting empirical risk minimization problem is susceptible to overfitting

if the number of training samples is small relative to the feature dimension. In order to combat overfitting, we propose to solve the DRO problem

$$J_N^\star = \min_{\theta \in \Theta} \ R^\star(\theta, \widehat{\mathbb{P}}_N^f), \tag{4}$$

which minimizes the worst-case risk over all distributions close to $\widehat{\mathbb{P}}_N^f$. Here, $R^\star$ is defined through

$$R^\star(\theta, \mathbb{P}') = \sup_{\mathbb{Q} \in \Pi} \left\{ R(\theta, \mathbb{Q}) : \mathrm{D}(\mathbb{P}' \| \mathbb{Q}) \le r \right\} \tag{5}$$

and thus evaluates the worst-case risk of a given parameter $\theta \in \Theta$ in view of all distributions $\mathbb{Q}$ that have a relative entropy distance of at most $r$ from a given nominal distribution $\mathbb{P}' \in \Pi$. In the remainder we use $J_N^\star$ and $\theta_N^\star$ to denote the minimum and a minimizer of problem (4), respectively.

**Main results.** The main theoretical results of this paper can be summarized as follows.

(i) *Out-of-sample guarantee.* We show that the optimal value of the DRO problem (4) provides an upper confidence bound on the risk of its optimal solution $\theta_N^\star$. Specifically, we prove that

$$\mathbb{P}^N \left( R(\theta_N^\star, \mathbb{P}^f) > J_N^\star \right) \le e^{-rN + o(N)}, \tag{6}$$

where $\mathbb{P}^f = f(\mathbb{P})$ is the I-projection of $\mathbb{P}$. If $\Xi$ is finite, then (6) can be strengthened to a finite sample bound that holds for every $N$ if the right hand side is replaced with $e^{-rN}(N+1)^{|\Xi|}$.

(ii) *Statistical efficiency.* In a sense to be made precise below, the DRO problem (4) provides the least conservative approximation for (1) whose solution satisfies the out-of-sample guarantee (6).

(iii) *Computational tractability.* We prove that the I-projection $\widehat{\mathbb{P}}_N^f$ can be computed via a regularized fast gradient method whenever one can efficiently project onto $E$. Given $\widehat{\mathbb{P}}_N^f$, we then show that $\theta_N^\star$ can be found by solving a tractable convex program whenever $\Theta$ is a convex and conic representable set, while $L(\theta, \xi)$ is a convex and conic representable function of $\theta$ for any fixed $\xi$.

## 4  Statistical guarantees

Throughout this section, we equip $\mathcal{P}(\Xi)$ with the topology of weak convergence. As $L(\theta, \xi)$ is continuous on $\Theta \times \Xi$ and $\Xi$ is compact, this implies that the risk $R(\theta, \mathbb{Q})$ is continuous on $\Theta \times \mathcal{P}(\Xi)$. The DRO problem (4) is constructed from the I-projection of the empirical distribution, which, in turn, is constructed from the given training samples. Thus, $\theta_N^\star$ constitutes a data-driven decision. Other data-driven decisions can be obtained by solving surrogate optimization problems of the form

$$\widehat{J}_N = \min_{\theta \in \Theta} \ \widehat{R}(\theta, \widehat{\mathbb{P}}_N^f), \tag{7}$$

where $\widehat{R} : \Theta \times \Pi \to \mathbb{R}$ is a continuous function that uses the empirical I-projection $\widehat{\mathbb{P}}_N^f$ to predict the true risk $R(\theta, \mathbb{P}^f)$ of $\theta$ under the true I-projection $\mathbb{P}^f$. From now on we thus refer to $\widehat{R}$ as a predictor, and we use $\widehat{J}_N$ and $\widehat{\theta}_N$ to denote the minimum and a minimizer of problem (7), respectively. We call a predictor $\widehat{R}$ *admissible* if $\widehat{J}_N$ provides an upper confidence bound on the risk of $\widehat{\theta}_N$ in the sense that

$$\limsup_{N \to \infty} \frac{1}{N} \log \mathbb{P}^N \left( R(\widehat{\theta}_N, \mathbb{P}^f) > \widehat{J}_N \right) \le -r \quad \forall \mathbb{P} \in \mathcal{P}(\Xi) \tag{8}$$

for some prescribed $r > 0$. The inequality (8) requires the true risk of the minimizer $\widehat{\theta}_N$ to exceed the optimal value $\widehat{J}_N$ of the surrogate optimization problem (7) with a probability that decays exponentially at rate $r$ as the number $N$ of training samples tends to infinity. The following theorem asserts that the DRO predictor $R^\star$ defined in (5), which evaluates the worst-case risk of any given $\theta$ across a relative entropy ball of radius $r$, almost satisfies (8) and is thus essentially admissible.

**Theorem 4.1** (Out-of-sample guarantee). *If $r > 0$, $0 \in \mathrm{int}(E)$ and for every $z \in \mathbb{R}^d$ there exists an uncertainty realization $\xi \in \Xi$ such that $z^\top \psi(\xi) > 0$, then the DRO predictor $R^\star$ defined in (5) is continuous on $\Theta \times \Pi$. In addition, $\widehat{R} = R^\star + \varepsilon$ is an admissible data-driven predictor for every $\varepsilon > 0$.*

Theorem 4.1 implies that, for any fixed $\varepsilon > 0$, the DRO predictor $R^\star$ provides an upper confidence bound $J_N^\star + \varepsilon$ on the true risk $R(\theta_N^\star, \mathbb{P}^f)$ of the data-driven decision $\theta_N^\star$ that becomes increasingly reliable as $N$ grows. Of course, the reliability of *any* upper confidence bound trivially improves if it is increased. Finding *some* upper confidence bound is thus easy. The next theorem shows that the DRO predictor actually provides the *best possible* (asymptotically smallest) upper confidence bound.

**Theorem 4.2** (Statistical efficiency). *Assume that all conditions of Theorem 4.1 hold. If $J_N^\star$ and $R^\star$ are defined as in (4) and (5), while $\widehat{J}_N$ is defined as in (7) for any admissible data-driven predictor $\widehat{R}$, then we have $\lim_{N\to\infty} J_N^\star \leq \lim_{N\to\infty} \widehat{J}_N$ $\mathbb{P}^\infty$-almost surely irrespective of $\mathbb{P} \in \mathcal{P}(\Xi)$.*

One readily verifies that the limits in Theorem 4.2 exist. Indeed, if $\widehat{R}$ is an arbitrary data-driven predictor, then the optimal value $\widehat{J}_N$ of the corresponding surrogate optimization problem converges $\mathbb{P}$-almost surely to $\min_{\theta \in \Theta} \widehat{R}(\theta, \mathbb{P}^f)$ as $N$ tends infinity provided that the training samples are drawn independently from $\mathbb{P}$. This is a direct consequence of the following three observations. First, the optimal value function $\min_{\theta \in \Theta} \widehat{R}(\theta, \mathbb{P}^f)$ is continuous in $\mathbb{P}^f \in \Pi$ thanks to Berge's maximum theorem [7, pp. 115–116], which applies because $\widehat{R}$ is continuous and $\Theta$ is compact. Second, the I-projection $\mathbb{P}^f = f(\mathbb{P})$ is continuous in $\mathbb{P} \in \mathcal{P}(\Xi)$ thanks to [70, Theorem 9.17], which applies because the relative entropy is strictly convex in its first argument [21, Lemma 6.2.12]. Third, the strong law of large numbers implies that the empirical distribution $\widehat{\mathbb{P}}_N$ converges weakly to the data-generating distribution $\mathbb{P}$ as the sample size $N$ grows. Therefore, we have

$$\lim_{N\to\infty} \widehat{J}_N = \lim_{N\to\infty} \min_{\theta \in \Theta} \widehat{R}\left(\theta, f(\widehat{\mathbb{P}}_N)\right) = \min_{\theta \in \Theta} \widehat{R}\left(\theta, f\left(\lim_{N\to\infty} \widehat{\mathbb{P}}_N\right)\right) = \min_{\theta \in \Theta} \widehat{R}(\theta, \mathbb{P}^f) \quad \mathbb{P}\text{-a.s.}$$

In summary, Theorems 4.1 and 4.2 assert that the DRO predictor $R^\star$ is (essentially) admissible and that it is the least conservative of all admissible data-driven predictors, respectively. Put differently, the DRO predictor makes the most efficient use of the available data among all data-driven predictors that offer the same out-of-sample guarantee (8). In the special case when $\Xi$ is finite, the asymptotic out-of-sample guarantee (8) can be strengthened to a finite sample guarantee that holds for every $N \in \mathbb{N}$.

**Corollary 4.3** (Finite sample guarantee). *If $R^\star$ is defined as in (5), then*

$$\frac{1}{N} \log \mathbb{P}^N \left(R^\star(\theta_N^\star, \mathbb{P}^f) > J_N^\star\right) \leq \frac{\log(N+1)}{N}|\Xi| - r \quad \forall N \in \mathbb{N}. \tag{9}$$

We now temporarily use $R_r^\star$ to denote the DRO predictor defined in (5), which makes its dependence on $r$ explicit. Note that if $r > 0$ is kept constant, then $R_r^\star(\theta, \widehat{\mathbb{P}}_N^f)$ is neither an unbiased nor a consistent estimator for $R(\theta, \mathbb{P}^f)$. Consistency can be enforced, however, by shrinking $r$ as $N$ grows.

**Theorem 4.4** (Asymptotic consistency). *Let the assumptions of Proposition 3.4 hold and $\{r_N\}_{N\in\mathbb{N}}$ be a sequence of non-negative reals with $\lim_{N\to\infty} r_N = 0$. If the loss function $L(\theta, \xi)$ is Lipschitz continuous in $\xi$ with Lipschitz constant $\Lambda > 0$ uniformly across all $\theta \in \Theta$, then we have*

$$\lim_{N\to\infty} R_{r_N}^\star(\theta, \widehat{\mathbb{P}}_N^f) = R(\theta, \mathbb{P}^f) \quad \mathbb{P}^\infty\text{-a.s. } \forall \theta \in \Theta, \tag{10a}$$

$$\lim_{N\to\infty} \min_{\theta \in \Theta} R_{r_N}^\star(\theta, \widehat{\mathbb{P}}_N^f) = \min_{\theta \in \Theta} R(\theta, \mathbb{P}^f) \quad \mathbb{P}^\infty\text{-a.s.} \tag{10b}$$

**Remark 4.5** (Choice of radius). *Theorem 4.2 shows that the ambiguity set used in our paper displays a strong Pareto-optimality property, i.e., it leads to the least conservative predictor, uniformly across all estimator realizations, for which the out-of-sample disappointment probability is guaranteed to decay exponentially at rate $r$. Therefore, the radius $r$ has a direct operational interpretation that captures the risk tolerance of the decision maker—it is chosen subjectively. Since the statistical guarantees of Theorem 4.1 are asymptotic, selecting the radius $r$ when we only have access to finitely many samples is challenging, and in practice $r$ is usually selected via cross validation.*

We now exemplify our DRO approach and its statistical guarantees in the context of the off-policy evaluation problem introduced in Section 3.

**Example 4.6** (Off-policy evaluation). *Consider again the OPE problem introduced in Example 3.6. We now aim to construct an estimator for the performance of the evaluation policy $V_{\pi_e} = \mathbb{E}_{f(\mathbb{P})}[\xi]$ based on the available behavioral policy and its empirical cost. As described in Example 3.6, we choose $\Pi$ such that $\mu_e \circ c^{-1} = f(\mathbb{P})$, where $\mathbb{P} = \mu_b \circ c^{-1} \in \mathcal{P}(\Xi)$. Given the behavioral data $(\widehat{s}_i, \widehat{a}_i) \sim \mu_b$ for $i = 1, \dots N$, we then construct the empirical distribution $\widehat{\mathbb{P}}_N = \frac{1}{N}\sum_{i=1}^N \delta_{c(\widehat{s}_i, \widehat{a}_i)}$. Our statistical results require the samples $(\widehat{s}_i, \widehat{a}_i)$ to be i.i.d., which can be enforced approximately by discarding a sufficient number of intermediate samples, for example. We emphasize, however, that the proposed large deviation framework readily generalizes to situations in which there is a single trajectory of correlated data [39, 72]. Details are omitted for brevity. The value function $V_{\pi_e}$*

*under the evaluation policy can now be approximated by $J_N^\star = R^\star(\widehat{\mathbb{P}}_N^f)$, where $R^\star$ denotes the DRO predictor (5). As $\Xi$ is finite, Corollary 4.3 provides the generalization bound*

$$\mathbb{P}^N\left(V_{\pi_e} \le J_N^\star\right) \ge 1 - (N+1)^{|\mathcal{S}|+|\mathcal{A}|}e^{-rN} \quad \forall \mathbb{P} \in \mathcal{P}(\Xi), \tag{11}$$

*which holds for all $N \in \mathbb{N}$.*

## 5 Efficient computation

We now outline an efficient procedure to solve the DRO problem (4). This procedure consists of two steps. First, we propose an algorithm to compute the I-projection $\widehat{\mathbb{P}}_N^f = f(\widehat{\mathbb{P}}_N)$ of the empirical distribution $\widehat{\mathbb{P}}_N$ corresponding to the training samples $\widehat{\xi}_1, \ldots, \widehat{\xi}_N$. Given $\widehat{\mathbb{P}}_N^f$, we then show how to compute the worst-case risk $R^\star(\theta, \widehat{\mathbb{P}}_N^f)$ and a corresponding optimizer $\theta_N^\star$ over the search space $\Theta$.

**Computation of the I-projection.** Computing the I-projection of the empirical distribution $\widehat{\mathbb{P}}_N$ is a non-trivial task because it requires solving the infinite-dimensional optimization problem (3). Generally, one would expect that the difficulty of evaluating $f(\widehat{\mathbb{P}}_N)$ depends on the structure of the set $\Pi$, which is encoded by $\psi$ and $E$; see (2). Thanks to the discrete nature of the empirical distribution $\widehat{\mathbb{P}}_N$, however, we can leverage recent advances in convex optimization together with an algorithm proposed in [71] to show that $f(\widehat{\mathbb{P}}_N)$ can be evaluated efficiently for a large class of sets $\Pi$.

In the following we let $\eta = (\eta_1, \eta_2)$ be a smoothing parameter with $\eta_1, \eta_2 > 0$, and we let $L_\eta > 0$ be a learning rate that may depend on $\eta$. In addition, we denote by $z \in \mathbb{R}^d$ the vector of dual variables of the constraint $\mathbb{E}_{\mathbb{Q}}[\psi(\xi)] \in E$ in problem (3), and we define $G_\eta : \mathbb{R}^d \to \mathbb{R}^d$ with

$$G_\eta(z) = -\pi_E(\eta_1^{-1}z) - \eta_2 z + \frac{\sum_{i=1}^N \psi(\widehat{\xi}_i) \exp\left(-\sum_{j=1}^d z_j\,\psi_j(\widehat{\xi}_i)\right)}{\sum_{i=1}^N \exp\left(-\sum_{j=1}^d z_j\,\psi_j(\widehat{\xi}_i)\right)} \tag{12}$$

as a smoothed gradient of the dual objective, where $\pi_E$ denotes the projection operator onto $E$ defined through $\pi_E(z) = \arg\min_{x \in E} \|x - z\|_2^2$. The corresponding smoothed dual of the I-projection problem (3) can then be solved with the fast gradient method described in Algorithm 1. The complexity of evaluating $G_\eta$, and thus the per-iteration complexity of Algorithm 1, is determined by the projection operator onto $E$. For simple sets (e.g., 2-norm balls or hybercubes) the solution is available in closed form, and for many other sets (e.g., simplices or 1-norm balls) it can be computed cheaply, see [53, Section 5.4] for a comprehensive survey.

---

**Algorithm 1:** Fast gradient method for smooth & strongly convex optimization [47]

---

Choose $w_0 = z_0 \in \mathbb{R}^d$ and $\eta \in \mathbb{R}_{++}^2$

**For** $k \in \mathbb{N}$          **Step 1:**    Set $z_{k+1} = w_k + \frac{1}{L_\eta}G_\eta(w_k)$

                               **Step 2:**    Compute $w_{k+1} = z_{k+1} + \frac{\sqrt{L_\eta}-\sqrt{\eta_2}}{\sqrt{L_\eta}+\sqrt{\eta_2}}(z_{k+1} - z_k)$

---

Any output $z_k$ of Algorithm 1 after $k$ iterations can be used to construct a candidate solution

$$\widehat{\mathbb{Q}}_k = \frac{\sum_{j=1}^N \exp\left(-\sum_{i=1}^d (z_k)_i \psi_i(\widehat{\xi}_j)\right)\delta_{\widehat{\xi}_j}}{\sum_{j=1}^N \exp\left(-\sum_{i=1}^d (z_k)_i \psi_i(\widehat{\xi}_j)\right)} \tag{13}$$

for problem (3) that approximates the I-projection $\widehat{\mathbb{P}}_N^f$. The convergence guarantees for Algorithm 1 and, in particular, the approximation quality of (13) with respect to $\widehat{\mathbb{P}}_N^f$ detailed in Theorem 5.2 below require that problem (3) admits a Slater point $\mathbb{P}^\circ$ in the sense of the following assumption.

**Assumption 5.1** (Slater point). *Problem (3) admits a Slater point $\mathbb{P}^\circ \in \Pi$ that satisfies*

$$\delta = \min_{y \notin E} \|\mathbb{E}_{\mathbb{P}^\circ}[\psi(\xi)] - y\|_2 > 0.$$

Finding a Slater point $\mathbb{P}^\circ$ may be difficult in general. However, $\mathbb{P}^\circ$ can be constructed systematically if $\psi$ is a polynomial [71, Remark 8], for example. Given $\mathbb{P}^\circ$ and a tolerance $\varepsilon > 0$, we then define

$$
\begin{aligned}
&C = \mathsf{D}(\mathbb{P}^\circ \| \widehat{\mathbb{P}}_N), \qquad D = \tfrac{1}{2}\max_{y \in E}\|y\|_2, \qquad \eta_1 = \tfrac{\varepsilon}{4D}, \qquad \eta_2 = \tfrac{\varepsilon\delta^2}{2C^2}, \\
&\alpha = \max_{\xi \in \Xi}\|\psi(\xi)\|_\infty, \quad L_\eta = 1/\eta_1 + \eta_2 + (\max_{\xi \in \Xi}\|\psi(\xi)\|_\infty)^2, \\
&M_1(\varepsilon) = 2\left(\sqrt{\tfrac{8DC^2}{\varepsilon^2\delta^2} + \tfrac{2\alpha^2 C^2}{\varepsilon\delta^2} + 1}\right)\log\left(\tfrac{10(\varepsilon + 2C)}{\varepsilon}\right), \\
&M_2(\varepsilon) = 2\left(\sqrt{\tfrac{8DC^2}{\varepsilon^2\delta^2} + \tfrac{2\alpha^2 C^2}{\varepsilon\delta^2} + 1}\right)\log\left(\tfrac{C}{\varepsilon\delta(2 - \sqrt{3})}\sqrt{4\left(\tfrac{4D}{\varepsilon} + \alpha^2 + \tfrac{\varepsilon\delta^2}{2C^2}\right)\left(C + \tfrac{\varepsilon}{2}\right)}\right).
\end{aligned}
\tag{14}
$$

**Theorem 5.2** (Almost linear convergence rate). *If Assumption 5.1 holds and $\varepsilon > 0$, then the candidate solution* (13) *obtained after $k = \lceil \max\{M_1(\varepsilon), M_2(\varepsilon)\}\rceil$ iterations of Algorithm 1 satisfies*

$$
\text{Optimality:} \qquad |\mathsf{D}(\widehat{\mathbb{Q}}_k \| \widehat{\mathbb{P}}_N) - \mathsf{D}(\widehat{\mathbb{P}}_N^f \| \widehat{\mathbb{P}}_N)| \le 2(1 + 2\sqrt{3})\varepsilon,
\tag{15a}
$$

$$
\text{Feasibility:} \qquad \mathsf{d}\left(\mathbb{E}_{\widehat{\mathbb{Q}}_k}[\psi(\xi)], E\right) \le \tfrac{2\varepsilon\delta}{C},
\tag{15b}
$$

*where we use the definitions* (14)*, and the function $\mathsf{d}(\cdot, E)$ denotes the Euclidean distance to the set $E$ defined through $\mathsf{d}(x, E) = \min_{y \in E}\|x - y\|_2$.*

Theorem 5.2 implies that Algorithm 1 needs at most $O(\tfrac{1}{\varepsilon}\log\tfrac{1}{\varepsilon})$ iterations to find an $O(\varepsilon)$-suboptimal and $O(\varepsilon)$-feasible solution for the I-projection problem (3). These results are derived via convex programming and duality by using the double smoothing techniques introduced in [22] and [71].

**Computation of the DRO predictor.** Equipped with Algorithm 1 to efficiently approximate $\widehat{\mathbb{P}}_N^f$ via $\widehat{\mathbb{Q}}_k$, the DRO predictor $R^\star(\theta, \widehat{\mathbb{P}}_N^f)$ defined in (4) can be approximated by $R^\star(\theta, \widehat{\mathbb{Q}}_k)$ because the function $R^\star$ is continuous. We now show that the worst-case risk evaluation problem (5) admits a dual representation, which generalizes [77, Proposition 5].

**Proposition 5.3** (Dual representation of $R^\star$). *If $r > 0$, then the DRO predictor $R^\star$ satisfies*

$$
R^\star(\theta, \mathbb{P}') = \begin{cases} \displaystyle\inf_{\alpha \in \mathbb{R}, z \in \mathbb{R}^d} & \alpha + \sigma_E(z) - e^{-r}\exp\left(\mathbb{E}_{\mathbb{P}'}[\log(\alpha - L(\theta, \xi) + z^\top \psi(\xi))]\right) \\ \text{s.t.} & \alpha \ge \max_{\xi \in \Xi} L(\theta, \xi) - z^\top \psi(\xi) \end{cases}
\tag{16}
$$

*for ever $\theta \in \Theta$ and $\mathbb{P}' \in \Pi$, where $\sigma_E(z) = \max_{x \in E} x^\top z$ denotes the support function of $E$.*

Proposition 5.3 implies that if $L(\theta, \xi)$ is convex in $\theta$ for every $\xi$, then the DRO predictor (5) coincides with the optimal value of a finite-dimensional convex program. Note that the objective function of (16) can be evaluated cheaply whenever the support function of $E$ is easy to compute and $\mathbb{P}'$ has finite support (e.g., if $\mathbb{P}'$ is set to an output $\widehat{\mathbb{Q}}_k$ of Algorithm 1). In addition, the robust constraint in (16) can be expressed in terms of explicit convex constraints if $L$, $\Xi$ and $\psi$ satisfy certain regularity conditions. A trivial condition is that $\Xi$ is finite. More general conditions are described in [6].

# 6  Experimental results

We now assess the empirical performance of the MDI-DRO method in our two running examples.[1]

**Synthetic dataset — covariate shift adaptation.** The first two experiments revolve around the logistic regression problem with a distribution shift described in Example 3.2. Specifically, we consider a synthetic dataset where the test data is affected by a covariate shift, which constitutes a special case of a distribution shift. Detailed information about the data generation process is provided in Appendix 7.4. Our numerical experiments reveal that the proposed MDI-DRO method significantly outperforms the naive ERM method in the sense that its out-of-sample risk has both a lower mean as well as a lower variance; see Figures 1a and 1b. We also compare MDI-DRO against the IWERM method, which accounts for the distribution shift by assigning importance weights $p^\star(\cdot)/p(\cdot)$ to the training samples, where $p^\star(\cdot)$ and $p(\cdot)$ denote the densities of the test distribution $\mathbb{P}^\star$ and training distribution $\mathbb{P}$, respectively. These importance weights are assumed to be known in IWERM. In

---

[1]All simulations were implemented in MATLAB and run on a 4GHz CPU with 16Gb RAM. The Matlab code for reproducing the plots is available from `https://github.com/tobsutter/PMDI_DRO`.

contrast, MDI-DRO does *not* require any knowledge of the test distribution other than its membership in $\Pi$. Nevertheless, MDI-DRO displays a similar out-of-sample performance as IWERM even though it has less information about $\mathbb{P}^\star$, and it achieves a lower variance than IWERM; see Figures 1c-1d. Figure 1e shows how the reliability of the upper confidence bound $J_N^\star$ and the out-of-sample risk $R(\theta_N^\star, \mathbb{P}^\star)$ change with the regularization parameter $r$. Additional results are reported in Figure 4 in the appendix. These results confirm that small regularization parameters $r$ lead to small out-of-sample risk and that increasing $r$ improves the reliability of the upper confidence bound $J_N^\star$.

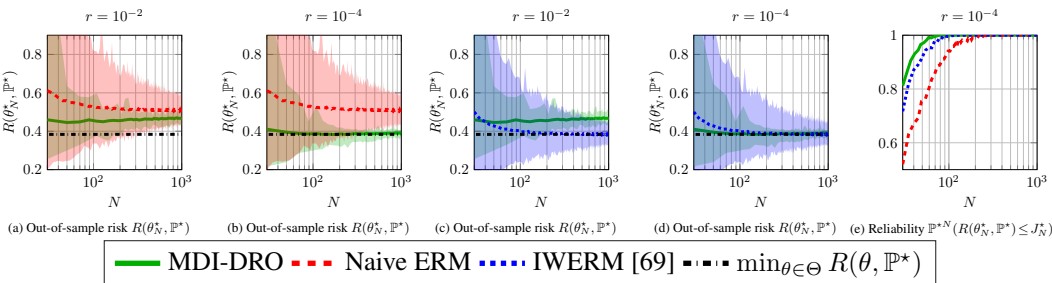

Figure 1: Results for a synthetic dataset with $m = 6$. Shaded areas and lines represent ranges and mean values across 1000 independent experiments, respectively.

**Real data — classification under sample bias.** The second experiment addresses the heart disease classification task of Example 3.2 based on a real dataset[2] consisting of $N^\star$ i.i.d. samples from an unknown test distribution $\mathbb{P}^\star$. To assess the effects of a distribution shift, we construct a biased training dataset $\{(\widehat{x}_1, \widehat{y}_1), \ldots, (\widehat{x}_N, \widehat{y}_N)\}$, $N < N^\star$, in which male patients older than 60 years are substantially over-represented. Specifically, the $N$ training samples are drawn randomly from the set of the 20% oldest male patients. Thus, the training data follows a distribution $\mathbb{P} \neq \mathbb{P}^\star$. Even though the test distribution $\mathbb{P}^\star$ is unknown, we assume to know the empirical mean $m = \frac{1}{N^\star} \sum_{i=1}^{N^\star} (\widehat{x}_i, \widehat{y}_i)$ of the entire dataset to within an absolute error $\Delta m > 0$. The test distribution thus belongs to the set $\Pi$ defined in (2) with $E = [m - \Delta m 1, m + \Delta m 1]$ and with $\psi(x, y) = (x, y)$. We compare the proposed MDI-DRO method for classification against the naive ERM method that ignores the sample bias. In addition, we use a logistic regression model trained on the entire dataset as an (unachievable) ideal benchmark. Figure 2a shows the out-of-sample cost, Figure 2b the upper confidence bound $J_N^\star$ and Figure 2c the misclassification rates of the different methods as the radius $r$ of the ambiguity set is swept. Perhaps surprisingly, for some values of $r$ the classification performance of MDI-DRO is comparable to that of the logistic regression method trained on the entire dataset.

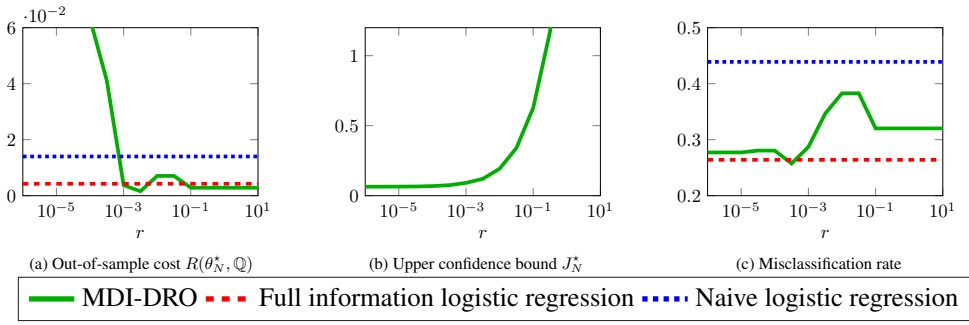

Figure 2: Heart disease classification example with $m = 6$, $N = 20$, $N^\star = 303$ and $\Delta m = 10^{-3}$.

**OPE for MDPs — inventory control.** We now consider the OPE problem of Examples 3.6 and 4.6. A popular estimator for the cost $V_{\pi_e}$ of the evaluation policy is the inverse propensity score (IPS) [57]

$$\widehat{J}_N^{\mathrm{IPS}} = \frac{1}{N} \sum_{i=1}^N c(\widehat{s}_i, \widehat{a}_i) \frac{\mu_e(\widehat{s}_i, \widehat{a}_i)}{\mu_b(\widehat{s}_i, \widehat{a}_i)}.$$

Hoeffding's inequality then gives rise to the simple concentration bound

$$\mathbb{P}^N \left( V_{\pi_e} \leq \widehat{J}_N^{\mathrm{IPS}} + \varepsilon \right) \geq 1 - e^{\frac{-2N\varepsilon^2}{b^2}} \quad \forall \varepsilon > 0, \ \forall N \in \mathbb{N}, \tag{17}$$

---

[2]https://www.kaggle.com/ronitf/heart-disease-uci

where $b = \max_{s \in \mathcal{S}, a \in \mathcal{A}} c(s,a)\mu_{\mathsf{e}}(s,a)/\mu_{\mathsf{b}}(s,a)$. As $b$ is typically a large constant, the finite sample bound (11) for $J_N^\star$ is often more informative than (17). In addition, the variance of $\widehat{J}_N^{\mathrm{IPS}}$ grows exponentially with the sample size $N$ [15, 13, 65]. As a simple remedy, one can cap the importance weights beyond some threshold $\beta > 0$ and construct the modified IPS estimator as

$$\widehat{J}_N^{\mathrm{IPS}_\beta} = \tfrac{1}{N} \sum_{i=1}^N c(\widehat{s}_i, \widehat{a}_i) \min\left\{\beta, \tfrac{\mu_{\mathsf{e}}(\widehat{s}_i, \widehat{a}_i)}{\mu_{\mathsf{b}}(\widehat{s}_i, \widehat{a}_i)}\right\}.$$

Decreasing $\beta$ reduces the variance of $\widehat{J}_N^{\mathrm{IPS}_\beta}$ but increases its bias. An alternative estimator for $V_{\pi_{\mathsf{e}}}$ is the doubly robust (DR) estimator $\widehat{J}_N^{\mathrm{DR}}$, which uses a control variate to reduce the variance of the IPS estimator. The DR estimator was first developed for contextual bandits [24] and then generalized to MDPs [32, 75]. We evaluate the performance of the proposed MDI-DRO estimator on a classical inventory control problem. A detailed problem description is relegated to Appendix 7.4. We sample both the evaluation policy $\pi_{\mathsf{e}}$ and the behavioral policy $\pi_{\mathsf{b}}$ from the uniform distribution on the space of stationary policies. The decision maker then has access to the evaluation policy $\pi_{\mathsf{e}}$ and to a sequence of i.i.d. state action pairs $\{\widehat{s}_i, \widehat{a}_i\}_{i=1}^N$ sampled from $\mu_{\mathsf{b}}$ as well as the observed empirical costs $\{\widehat{c}_i\}_{i=1}^N$, where $\widehat{c}_i = c(\widehat{s}_i, \widehat{a}_i)$. Figure 3 compares the proposed MDI-DRO estimator against the original and modified IPS estimators, the DR estimator and the ground truth expected cost of the evaluation policy. Figures 3a and 3b show that for small radii $r$, the MDI-DRO estimator outperforms the IPS estimators both in terms of accuracy and precision. Figure 3c displays the disappointment probabilities $\mathbb{P}^N(V_{\pi_{\mathsf{e}}} > \widehat{J}_N)$ analyzed in Theorem 4.1, where $\widehat{J}_N$ denotes any of the tested estimators.

**Acknowledgments.** We thank Mengmeng Li for helpful discussions. This research was supported by the Swiss National Science Foundation under the NCCR Automation, grant agreement 51NF40_180545.

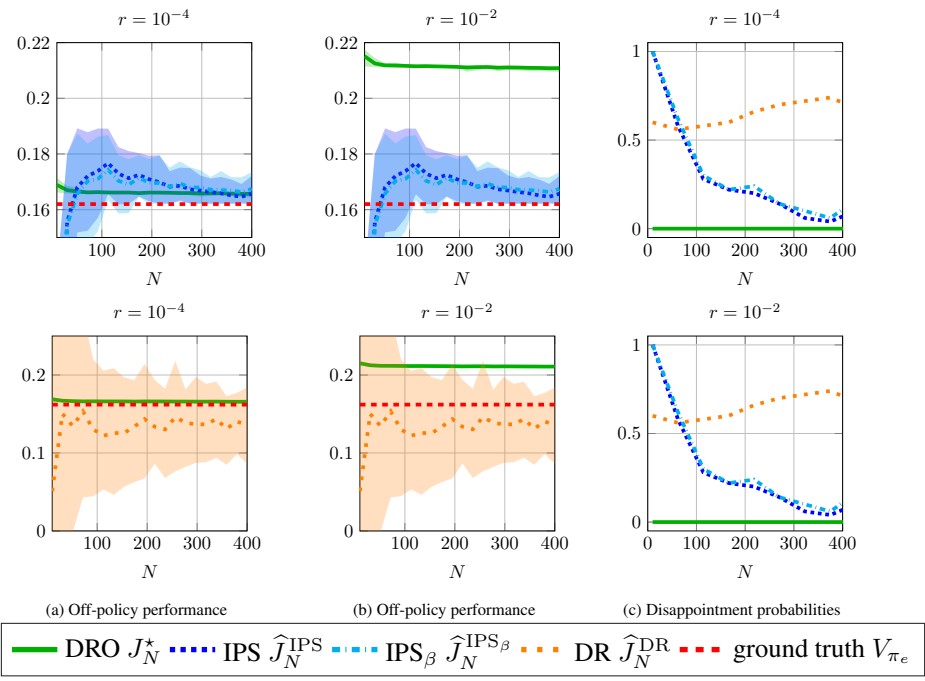

(a) Off-policy performance     (b) Off-policy performance     (c) Disappointment probabilities

DRO $J_N^\star$    IPS $\widehat{J}_N^{\mathrm{IPS}}$    IPS$_\beta$ $\widehat{J}_N^{\mathrm{IPS}_\beta}$    DR $\widehat{J}_N^{\mathrm{DR}}$    ground truth $V_{\pi_e}$

Figure 3: Shaded areas and lines represent 90% confidence intervals and mean values across 1000 independent experiments, respectively.

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
