| —— MDI-DRO    ▪▪▪ Naive ERM    ▪▪▪▪ IWERM [69]    ▪-▪-▪ $\min_{\theta \in \Theta} R(\theta, \mathbb{P}^\star)$ |

Figure 1: Results for a synthetic dataset with $m = 6$. Shaded areas and lines represent ranges and mean values across 1000 independent experiments, respectively.

**Real data — classification under sample bias.** The second experiment addresses the heart disease classification task of Example 3.2 based on a real dataset[2] consisting of $N^\star$ i.i.d. samples from an unknown test distribution $\mathbb{P}^\star$. To assess the effects of a distribution shift, we construct a biased training dataset $\{(\widehat{x}_1, \widehat{y}_1), \ldots, (\widehat{x}_N, \widehat{y}_N)\}$, $N < N^\star$, in which male patients older than 60 years are substantially over-represented. Specifically, the $N$ training samples are drawn randomly from the set of the 20% oldest male patients. Thus, the training data follows a distribution $\mathbb{P} \neq \mathbb{P}^\star$. Even though the test distribution $\mathbb{P}^\star$ is unknown, we assume to know the empirical mean $m = \frac{1}{N^\star} \sum_{i=1}^{N^\star} (\widehat{x}_i, \widehat{y}_i)$ of the entire dataset to within an absolute error $\Delta m > 0$. The test distribution thus belongs to the set $\Pi$ defined in (2) with $E = [m - \Delta m \mathbf{1}, m + \Delta m \mathbf{1}]$ and with $\psi(x, y) = (x, y)$. We compare the proposed MDI-DRO method for classification against the naive ERM method that ignores the sample bias. In addition, we use a logistic regression model trained on the entire dataset as an (unachievable) ideal benchmark. Figure 2a shows the out-of-sample cost, Figure 2b the upper confidence bound $J_N^\star$ and Figure 2c the misclassification rates of the different methods as the radius $r$ of the ambiguity set is swept. Perhaps surprisingly, for some values of $r$ the classification performance of MDI-DRO is comparable to that of the logistic regression method trained on the entire dataset.

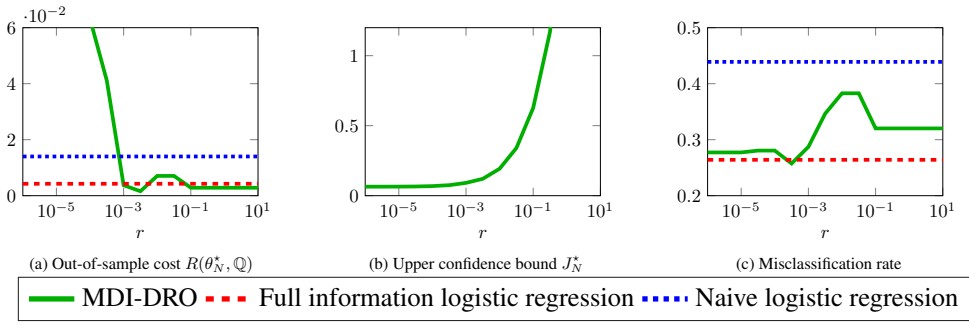

(a) Out-of-sample cost $R(\theta_N^\star, \mathbb{Q})$     (b) Upper confidence bound $J_N^\star$     (c) Misclassification rate

| —— MDI-DRO    ▪▪▪ Full information logistic regression    ▪▪▪▪ Naive logistic regression |

Figure 2: Heart disease classification example with $m = 6$, $N = 20$, $N^\star = 303$ and $\Delta m = 10^{-3}$.

**OPE for MDPs — inventory control.** We now consider the OPE problem of Examples 3.6 and 4.6. A popular estimator for the cost $V_{\pi_e}$ of the evaluation policy is the inverse propensity score (IPS) [57]
$$\widehat{J}_N^{\mathrm{IPS}} = \tfrac{1}{N} \sum_{i=1}^N c(\widehat{s}_i, \widehat{a}_i) \tfrac{\mu_e(\widehat{s}_i, \widehat{a}_i)}{\mu_b(\widehat{s}_i, \widehat{a}_i)}.$$
Hoeffding's inequality then gives rise to the simple concentration bound
$$\mathbb{P}^N\left(V_{\pi_e} \leq \widehat{J}_N^{\mathrm{IPS}} + \varepsilon\right) \geq 1 - e^{\frac{-2N\varepsilon^2}{b^2}} \quad \forall \varepsilon > 0, \ \forall N \in \mathbb{N}, \tag{17}$$

[2] https://www.kaggle.com/ronitf/heart-disease-uci

where $b = \max_{s \in \mathcal{S}, a \in \mathcal{A}} c(s,a)\mu_{\mathsf{e}}(s,a)/\mu_{\mathsf{b}}(s,a)$. As $b$ is typically a large constant, the finite sample bound (11) for $J_N^\star$ is often more informative than (17). In addition, the variance of $\widehat{J}_N^{\mathrm{IPS}}$ grows exponentially with the sample size $N$ [15, 13, 65]. As a simple remedy, one can cap the importance weights beyond some threshold $\beta > 0$ and construct the modified IPS estimator as

$$\widehat{J}_N^{\mathrm{IPS}_\beta} = \tfrac{1}{N} \sum_{i=1}^N c(\widehat{s}_i, \widehat{a}_i) \min\left\{ \beta, \tfrac{\mu_{\mathsf{e}}(\widehat{s}_i, \widehat{a}_i)}{\mu_{\mathsf{b}}(\widehat{s}_i, \widehat{a}_i)} \right\}.$$

Decreasing $\beta$ reduces the variance of $\widehat{J}_N^{\mathrm{IPS}_\beta}$ but increases its bias. An alternative estimator for $V_{\pi_{\mathsf{e}}}$ is the doubly robust (DR) estimator $\widehat{J}_N^{\mathrm{DR}}$, which uses a control variate to reduce the variance of the IPS estimator. The DR estimator was first developed for contextual bandits [24] and then generalized to MDPs [32, 75]. We evaluate the performance of the proposed MDI-DRO estimator on a classical inventory control problem. A detailed problem description is relegated to Appendix 7.4. We sample both the evaluation policy $\pi_{\mathsf{e}}$ and the behavioral policy $\pi_{\mathsf{b}}$ from the uniform distribution on the space of stationary policies. The decision maker then has access to the evaluation policy $\pi_{\mathsf{e}}$ and to a sequence of i.i.d. state action pairs $\{\widehat{s}_i, \widehat{a}_i\}_{i=1}^N$ sampled from $\mu_{\mathsf{b}}$ as well as the observed empirical costs $\{\widehat{c}_i\}_{i=1}^N$, where $\widehat{c}_i = c(\widehat{s}_i, \widehat{a}_i)$. Figure 3 compares the proposed MDI-DRO estimator against the original and modified IPS estimators, the DR estimator and the ground truth expected cost of the evaluation policy. Figures 3a and 3b show that for small radii $r$, the MDI-DRO estimator outperforms the IPS estimators both in terms of accuracy and precision. Figure 3c displays the disappointment probabilities $\mathbb{P}^N(V_{\pi_{\mathsf{e}}} > \widehat{J}_N)$ analyzed in Theorem 4.1, where $\widehat{J}_N$ denotes any of the tested estimators.

**Acknowledgments.** We thank Mengmeng Li for helpful discussions. This research was supported by the Swiss National Science Foundation under the NCCR Automation, grant agreement 51NF40_180545.

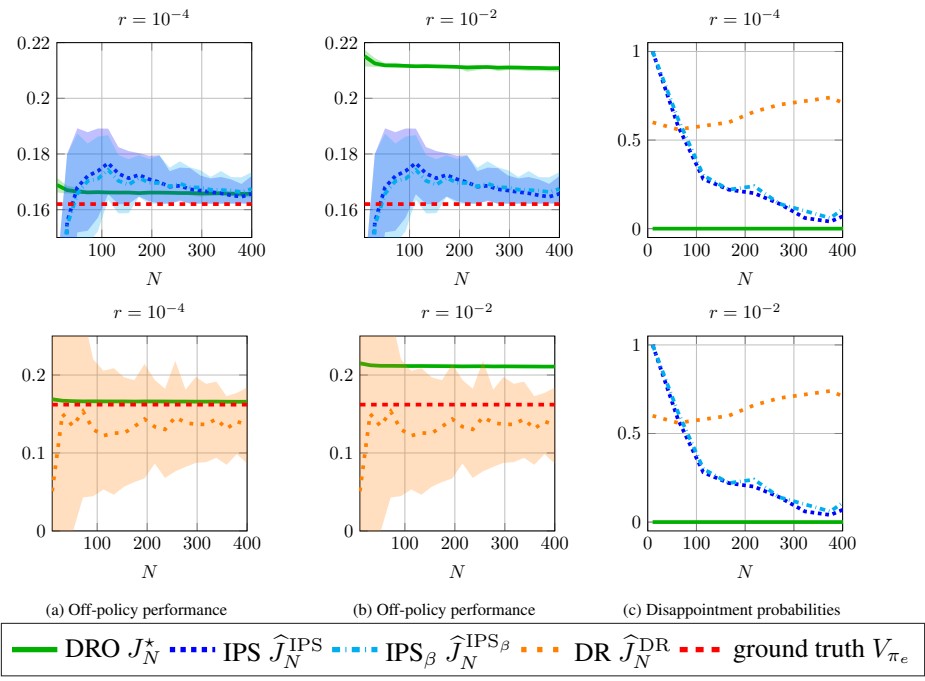

Figure 3: Shaded areas and lines represent 90% confidence intervals and mean values across 1000 independent experiments, respectively.

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

# 7 Appendix

The appendix details all proofs and provides some auxiliary results grouped by section.

## 7.1 Proofs of Section 3

*Proof of Proposition 3.4.* Denote by $\mathbb{P}_{\xi_1|\Pi}^N$ the probability distribution of $\xi_1$ with respect to $\mathbb{P}^N$ conditional on the event $\widehat{\mathbb{P}}_N \in \Pi$. By [18, Theorem 4], we then have

$$\lim_{N\to\infty} \mathsf{D}(\mathbb{P}_{\xi_1|\Pi}^N \| \mathbb{P}^f) = 0,$$

i.e., the conditional distribution $\mathbb{P}_{\xi_1|\Pi}^N$ converges in information to $\mathbb{P}^f$. As the moment-generating function $\mathbb{E}_{\mathbb{P}^f}[e^{tL(\theta,\xi)}]$ is finite for all $t$ in a neighborhood of 0, [17, Lemma 3.1] ensures that

$$\lim_{N\to\infty} \mathbb{E}_{\mathbb{P}^N}[L(\theta,\xi_1)|\widehat{\mathbb{P}}_N \in \Pi] = \lim_{N\to\infty} \mathbb{E}_{\mathbb{P}_{\xi_1|\Pi}^N}[L(\theta,\xi_1)] = \mathbb{E}_{\mathbb{P}^\star}[L(\theta,\xi_1)].$$

Thus, the claim follows. $\qquad\square$

*Proof of Proposition 3.5.* Proposition 3.5 can be seen as a generalization of [16, Exercise 12.6]. To simplify notation, we define $\alpha = \mathsf{D}(\mathbb{Q}\|\mathbb{P})$. Then, we have

$$\min_{\bar{\mathbb{Q}}\in\Pi} \mathsf{D}(\bar{\mathbb{Q}}\|\mathbb{P}) = \min_{\bar{\mathbb{Q}}\in\mathcal{P}(\Xi)} \sup_{\lambda\in\mathbb{R}} \mathsf{D}(\bar{\mathbb{Q}}\|\mathbb{P}) - \lambda\left(\int_\Xi \log\left(\frac{\mathrm{d}\mathbb{Q}}{\mathrm{d}\mathbb{P}}\right)\mathrm{d}\bar{\mathbb{Q}} - \alpha\right) \tag{18a}$$

$$= \max_{\lambda\in\mathbb{R}} \min_{\bar{\mathbb{Q}}\in\mathcal{P}(\Xi)} \mathsf{D}(\bar{\mathbb{Q}}\|\mathbb{P}) - \lambda\int_\Xi \log\left(\frac{\mathrm{d}\mathbb{Q}}{\mathrm{d}\mathbb{P}}\right)\mathrm{d}\bar{\mathbb{Q}} + \lambda\alpha \tag{18b}$$

$$= \max_{\lambda\in\mathbb{R}} -\log\int_\Xi \left(\frac{\mathrm{d}\mathbb{Q}}{\mathrm{d}\mathbb{P}}\right)^\lambda \mathrm{d}\mathbb{P} + \lambda\alpha = \alpha, \tag{18c}$$

where (18a) holds by the definition of the set $\Pi$, and (18b) follows from Sion's minimax theorem. The latter applies because the relative entropy $\mathsf{D}(\bar{\mathbb{Q}}\|\mathbb{P})$ is convex in $\bar{\mathbb{Q}}$ and the distribution family $\mathcal{P}(\Xi)$ is convex and weakly compact thanks to the compactness of $\Xi$. Finally, (18c) holds because of [71, Lemma 2], which implies that the inner minimization problem in (18b) is uniquely solved by the probability distribution $\bar{\mathbb{Q}}_\lambda^\star \in \mathcal{P}(\Xi)$ defined through

$$\bar{\mathbb{Q}}_\lambda^\star(B) = \frac{\int_B e^{\lambda\log\left(\frac{\mathrm{d}\mathbb{Q}}{\mathrm{d}\mathbb{P}}\right)}\mathrm{d}\mathbb{P}}{\int_\Xi e^{\lambda\log\left(\frac{\mathrm{d}\mathbb{Q}}{\mathrm{d}\mathbb{P}}\right)}\mathrm{d}\mathbb{P}} = \frac{\int_B \left(\frac{\mathrm{d}\mathbb{Q}}{\mathrm{d}\mathbb{P}}\right)^\lambda \mathrm{d}\mathbb{P}}{\int_\Xi \left(\frac{\mathrm{d}\mathbb{Q}}{\mathrm{d}\mathbb{P}}\right)^\lambda \mathrm{d}\mathbb{P}} \quad \forall B \in \mathcal{B}(\Xi).$$

By inspecting the first-order optimality condition of the convex maximization problem in (18c) and remembering that $\alpha = \mathsf{D}(\mathbb{Q}\|\mathbb{P})$, one can then show that (18c) is solved by $\lambda^\star = 1$. The Nash equilibrium of the zero-sum game in (18b) is therefore given by $\lambda^\star$ and its unique best response $\bar{\mathbb{Q}}_{\lambda^\star}^\star = \mathbb{Q}$, and the solution $f(\mathbb{P})$ of the I-projection problem in (18a) coincides with $\mathbb{Q}$. $\quad\square$

## 7.2 Proofs of Section 4

*Proof of Theorem 4.1.* The continuity of $R^\star$ on $\Theta \times \Pi$ is established in Corollary 7.1 below.

In order to prove that the DRO predictor $R^\star$ is also admissible, we first prove that the following inequality holds for any fixed $\theta \in \Theta$ and $\mathbb{P} \in \mathcal{P}(\Xi)$.

$$\limsup_{N\to\infty} \frac{1}{N}\log\mathbb{P}^N\left(R(\theta,\mathbb{P}^f) > R^\star(\theta,\widehat{\mathbb{P}}_N^f)\right) \le -r \tag{19}$$

For the sake of concise notation, we then define the disappointment set

$$\mathcal{D}(\theta,\mathbb{P}) = \{\mathbb{P}' \in \mathcal{P}(\Xi) : R(\theta, f(\mathbb{P})) > R^\star(\theta, f(\mathbb{P}'))\}$$

containing all realizations $\mathbb{P}'$ of the empirical distribution $\widehat{\mathbb{P}}_N$, for which the true risk $R(\theta, f(\mathbb{P}))$ under the I-projection of the unknown true distribution exceeds the risk $R^\star(\theta, f(\mathbb{P}'))$ predicted by the distributionally robust predictor under the I-projection of the empirical distribution. Hence, $\bar{\mathcal{D}}(\theta,\mathbb{P})$

contains all realizations of $\widehat{\mathbb{P}}_N$ under which the distributionally robust predictor is too optimistic and thus leads to disappointment. Similarly, we define the weak disappointment set

$$\bar{\mathcal{D}}(\theta, \mathbb{P}) = \{\mathbb{P}' \in \mathcal{P}(\Xi) : R(\theta, f(\mathbb{P})) \geq R^\star(\theta, f(\mathbb{P}'))\},$$

which simply replaces the strict inequality in the definiton of $\bar{\mathcal{D}}(\theta, \mathbb{P})$ with a weak inequality. Recall now that $R^\star$ is continuous. In addition, note that $f$ is continuous thanks to [70, Theorem 9.17], which follows from the strict convexity of the relative entropy in its first argument [21, Lemma 6.2.12]. Therefore the set $\bar{\mathcal{D}}(\theta, \mathbb{P})$ is closed, and $\operatorname{cl} \mathcal{D}(\theta, \mathbb{P}) \subset \bar{\mathcal{D}}(\theta, \mathbb{P})$. The left hand side of (19) thus satisfies

$$\begin{aligned}
\limsup_{N\to\infty} \frac{1}{N} \log \mathbb{P}^N \left( R(\theta, f(\mathbb{P})) > R^\star(\theta, f(\widehat{\mathbb{P}}_N)) \right) &= \limsup_{N\to\infty} \frac{1}{N} \log \mathbb{P}^N \left( \widehat{\mathbb{P}}_N \in \mathcal{D}(\theta, \mathbb{P}) \right) \\
&\leq - \inf_{\mathbb{P}' \in \operatorname{cl} \mathcal{D}(\theta, \mathbb{P})} \mathsf{D}(\mathbb{P}' \| \mathbb{P}) \\
&\leq - \inf_{\mathbb{P}' \in \bar{\mathcal{D}}(\theta, \mathbb{P})} \mathsf{D}(\mathbb{P}' \| \mathbb{P}) \\
&\leq -r,
\end{aligned}$$

where the first inequality follows from Sanov's Theorem, which asserts that $\widehat{\mathbb{P}}_N$ satisfies a large deviation principle with the relative entropy as the rate function [21, Theorem 6.2.10]. The second inquality exploits the inclusion $\operatorname{cl} \mathcal{D}(\theta, \mathbb{P}) \subset \bar{\mathcal{D}}(\theta, \mathbb{P})$, and the last inequality holds because

$$\mathbb{P}' \in \bar{\mathcal{D}}(\theta, \mathbb{P}) \quad \implies \quad \mathsf{D}(f(\mathbb{P}') \| f(\mathbb{P})) \geq r \quad \implies \quad \mathsf{D}(\mathbb{P}' \| \mathbb{P}) \geq r,$$

where the first implication has been established in te proof of [77, Theorem 10], and the second implication follows from the data-processing inequality [19, Lemma 3.11]. This proves (19).

In the last step of the proof, we fix an arbitrary $\varepsilon > 0$ and show that

$$\limsup_{N\to\infty} \frac{1}{N} \log \mathbb{P} \left( R(\theta_N^\star, \mathbb{P}^f) > R^\star(\theta_N^\star, \widehat{\mathbb{P}}_N^f) + \varepsilon \right) \leq -r$$

for any $\mathbb{P} \in \mathcal{P}(\Xi)$, where $\theta_N^\star$ is defined as usual as a minimizer of (4). The proof of this generalized statement widely parallels that of [77, Theorem 11] and exploits the data processing inequality in a similar manner as in the proof of (19). Details are omitted for brevity. $\qquad\square$

*Proof of Theorem 4.2.* The proof is inspired by [77, Theorems 7 & 11]. We first show that any continuous admissible data-driven predictor $\widehat{R}$ satisfies the inequality

$$\lim_{N\to\infty} R^\star(\theta, f(\widehat{\mathbb{P}}_N)) \leq \lim_{N\to\infty} \widehat{R}(\theta, f(\widehat{\mathbb{P}}_N)) \quad \mathbb{P}^\infty\text{-a.s.} \tag{20}$$

for all $\theta \in \Theta$ and $\mathbb{P} \in \mathcal{P}(\Xi)$. As the empirical distribution $\widehat{\mathbb{P}}_N$ converges weakly to $\mathbb{P}$ and as $R^\star$, $\widehat{R}$ and $f$ represent continuous mappings, the inequality (20) is equivalent to

$$R^\star(\theta, f(\mathbb{P})) \leq \widehat{R}(\theta, f(\mathbb{P}))$$

for all $\theta \in \Theta$ and $\mathbb{P} \in \mathcal{P}(\Xi)$. Suppose now for the sake of contradiction there exists a continuous admissible predictor $\widehat{R}$, a parameter $\theta_0 \in \Theta$ and an asymptotic estimator realization $\mathbb{P}_0' \in \mathcal{P}(\Xi)$ with

$$\widehat{R}(\theta_0, f(\mathbb{P}_0')) < R^\star(\theta_0, f(\mathbb{P}_0')).$$

In fact, as $\widehat{R}$, $R^\star$ and $f$ are continuous functions, this strict inequality holds on a neighborhood of $\mathbb{P}_0'$. Next, define $\varepsilon = R^\star(\theta_0, f(\mathbb{P}_0')) - \widehat{R}(\theta_0, f(\mathbb{P}_0')) > 0$ and denote by $\bar{\mathbb{P}} \in \Pi$ an optimizer of the worst-case risk evaluation problem (5) for $\mathbb{P}' = f(\mathbb{P}_0')$, which satisfies $R^\star(\theta_0, f(\mathbb{P}_0')) = R(\theta_0, \bar{\mathbb{P}})$ and $\mathsf{D}(f(\mathbb{P}_0') \| \bar{\mathbb{P}}) \leq r$. By using a continuity argument as in the proof of [77, Theorem 10] and by exploiting the convexity of $\Pi$, one can then show that there exists a model $\mathbb{P}_0 \in \Pi$ with

$$R(\theta_0, \bar{\mathbb{P}}) < R(\theta, \mathbb{P}_0) + \varepsilon \quad \text{and} \quad \mathsf{D}(f(\mathbb{P}_0') \| \mathbb{P}_0) = r_0 < r. \tag{21}$$

All of this implies that

$$\widehat{R}(\theta_0, f(\mathbb{P}_0')) = R^\star(\theta_0, f(\mathbb{P}_0')) - \varepsilon = R(\theta_0, \bar{\mathbb{P}}) - \varepsilon < R(\theta_0, \mathbb{P}_0) = R(\theta_0, f(\mathbb{P}_0)), \tag{22}$$

where the three equalities follow from the definition of $\varepsilon$, the construction of $\bar{\mathbb{P}}$ and the observation that $f$ reduces to the identity mapping when restricted to $\Pi$. The inequality holds due to the first

relation in (21). In analogy to the proof of Theorem 4.1, we now introduce the disappointment set for the data-driven predictor $\widehat{R}$ under the data-generating distribution $\mathbb{P}_0$, that is,

$$\mathcal{D}(\theta_0, \mathbb{P}_0) = \left\{ \mathbb{P}' \in \mathcal{P}(\Xi) : R(\theta_0, f(\mathbb{P}_0)) > \widehat{R}(\theta_0, f(\mathbb{P}')) \right\}.$$

The relation (22) readily implies that $\mathbb{P}'_0 \in \mathcal{D}(\theta_0, \mathbb{P}_0)$. As the I-projection is idempotent (that is, $f \circ f = f$), one can further verify that $f(\mathbb{P}'_0) \in \mathcal{D}(\theta_0, \mathbb{P}_0)$. Denoting the empirical distribution of $N$ training samples drawn independently from $\mathbb{P}_0$ by $\widehat{\mathbb{P}}_{0,N}$, we thus find

$$
\begin{aligned}
\liminf_{N \to \infty} \frac{1}{N} \log \mathbb{P}_0^N \left( R(\theta_0, f(\mathbb{P}_0)) > \widehat{R}(\theta_0, f(\widehat{\mathbb{P}}_{0,N})) \right) &= \liminf_{N \to \infty} \frac{1}{N} \log \mathbb{P}_0^N \left( \widehat{\mathbb{P}}_{0,N} \in \mathcal{D}(\theta_0, \mathbb{P}_0) \right) \\
&\geq - \inf_{\mathbb{P}' \in \text{int}\, \mathcal{D}(\theta_0, \mathbb{P}_0)} \mathsf{D}(\mathbb{P}' \| \mathbb{P}_0) \\
&= - \inf_{\mathbb{P}' \in \mathcal{D}(\theta_0, \mathbb{P}_0)} \mathsf{D}(\mathbb{P}' \| \mathbb{P}_0) \\
&\geq - \mathsf{D}(f(\mathbb{P}'_0) \| \mathbb{P}_0) \\
&= -r_0 > -r,
\end{aligned}
$$

where the first inequality follows from Sanov's Theorem, which ensures that $\widehat{\mathbb{P}}_N$ satisfies a large deviation principle with the relative entropy as the rate function. The second equality holds because $\mathcal{D}(\theta_0, \mathbb{P}'_0)$ is open thanks to the continuity of $\widehat{R}$ and $f$, and the second inequality exploits our earlier insight that $f(\mathbb{P}'_0) \in \mathcal{D}(\theta_0, \mathbb{P}'_0)$. The last inequality, finally, follows from the second relation in (21). The above reasoning shows that $\widehat{R}$ fails to be admissible, and hence a data-driven predictor $\widehat{R}$ with the advertised properties cannot exist. Thus, $R^\star$ indeed satisfies the efficiency property (20).

To show that $\lim_{N \to \infty} J_N^\star \leq \lim_{N \to \infty} \widehat{J}_N$ $\mathbb{P}^\infty$-almost surely for all $\mathbb{P} \in \mathcal{P}(\Xi)$, we use (20) and adapt the proof of [77, Theorem 11] with obvious modifications. Details are omitted for brevity. $\quad\square$

*Proof of Corollary 4.3.* Recalling that Sanov's Theorem for finite state spaces offers finite sample bounds [16, Theorem 11.4.1], the claim can be established by repeating the proof of Theorem 4.1. $\quad\square$

*Proof of Theorem 4.4.* It suffices to prove (10b) because (10a) can be seen as a special case of (10b) when $\Theta = \{\theta\}$. In the remainder we denote by $\mathsf{d}_{\mathsf{TV}}(\mathbb{P}, \mathbb{Q})$ the total variation distance and by $\mathsf{d}_{\mathsf{W}_p}(\mathbb{P}, \mathbb{Q})$ the $p$-th Wasserstein distance ($p \in \mathbb{N}$) between two probability distributions $\mathbb{P}, \mathbb{Q} \in \mathcal{P}(\Xi)$. To make its dependence on the radius $r$ explicit, throughout this proof we temporarily use $R_r^\star$ to denote the DRO predictor (5). As usual, we use $\theta_N^\star \in \Theta$ to denote a minimizer of the DRO problem (4) with $\mathbb{P}' = \widehat{\mathbb{P}}_N^f$. In addition, we use $\widehat{\mathbb{Q}}_{N,\theta}^\star \in \Pi$ to denote a maximizer of the worst-case risk evaluation problem (5) with $\mathbb{P}' = \widehat{\mathbb{P}}_N^f$. By definition, this maximizer must satisfy the relations

$$R_{r_N}^\star(\theta, \widehat{\mathbb{P}}_N^f) = R(\theta, \widehat{\mathbb{Q}}_{N,\theta}^\star) \quad \text{and} \quad \mathsf{D}(\widehat{\mathbb{P}}_N^f \| \widehat{\mathbb{Q}}_{N,\theta}^\star) \leqslant r_N$$

for all $\theta \in \Theta$ and $N \in \mathbb{N}$. Pinsker's inequality then implies that

$$\sup_{\theta \in \Theta} \mathsf{d}_{\mathsf{TV}}\left(\widehat{\mathbb{P}}_N^f, \widehat{\mathbb{Q}}_{N,\theta}^\star\right) \leq \sup_{\theta \in \Theta} \sqrt{\frac{1}{2} \mathsf{D}(\widehat{\mathbb{P}}_N^f \| \widehat{\mathbb{Q}}_{N,\theta}^\star)} \leq \sqrt{\frac{r_N}{2}} \quad \forall N \in \mathbb{N}. \tag{23}$$

Thus, we find

$$
\begin{aligned}
\sup_{\theta \in \Theta} &\left\{ \left| R_{r_N}^\star(\theta, \widehat{\mathbb{P}}_N^f) - R(\theta, \mathbb{P}^f) \right| \right\} \\
&= \sup_{\theta \in \Theta} \left\{ \left| \mathbb{E}_{\widehat{\mathbb{Q}}_{N,\theta}^\star}[L(\theta, \xi)] - \mathbb{E}_{\mathbb{P}^f}[L(\theta, \xi)] \right| \right\} \\
&\leq \sup_{\theta \in \Theta} \left\{ \left| \mathbb{E}_{\widehat{\mathbb{Q}}_{N,\theta}^\star}[L(\theta, \xi)] - \mathbb{E}_{\widehat{\mathbb{P}}_N^f}[L(\theta, \xi)] \right| + \left| \mathbb{E}_{\widehat{\mathbb{P}}_N^f}[L(\theta, \xi)] - \mathbb{E}_{\mathbb{P}^f}[L(\theta, \xi)] \right| \right\} \\
&\leq \sup_{\theta \in \Theta} \left\{ \left| \mathbb{E}_{\widehat{\mathbb{Q}}_{N,\theta}^\star}[L(\theta, \xi)] - \mathbb{E}_{\widehat{\mathbb{P}}_N^f}[L(\theta, \xi)] \right| \right\} + \sup_{\theta \in \Theta} \left\{ \left| \mathbb{E}_{\widehat{\mathbb{P}}_N^f}[L(\theta, \xi)] - \mathbb{E}_{\mathbb{P}^f}[L(\theta, \xi)] \right| \right\} \\
&\leq \Lambda \sup_{\theta \in \Theta} \mathsf{d}_{\mathsf{W}_1}\left(\widehat{\mathbb{Q}}_{N,\theta}^\star, \widehat{\mathbb{P}}_N^f\right) + \Lambda \mathsf{d}_{\mathsf{W}_1}\left(\widehat{\mathbb{P}}_N^f, \mathbb{P}^f\right) \\
&\leq \Lambda C \sup_{\theta \in \Theta} \mathsf{d}_{\mathsf{TV}}\left(\widehat{\mathbb{Q}}_{N,\theta}^\star, \widehat{\mathbb{P}}_N^f\right) + \Lambda \mathsf{d}_{\mathsf{W}_2}\left(\widehat{\mathbb{P}}_N^f, \mathbb{P}^f\right),
\end{aligned}
$$

where the first three inequalities follow from the triangle inequality, the subadditivity of the supremum operator and the Kantorovich-Rubinstein theorem [79, Theorem 5.10], respectively. The last inequality holds because $\Xi$ is compact, which implies that the first Wasserstein distance can be bounded above by the total variation distance scaled with a positive constant $C$ [79, Theorem 6.15] and because $\mathsf{d}_{\mathsf{W}_1}(\cdot, \cdot) \leq \mathsf{d}_{\mathsf{W}_2}(\cdot, \cdot)$ thanks to Jensen's inequality. By (23), the first term in the above expression decays deterministically to zero as $N$ grows. The second term converges $\mathbb{P}^\infty$-almost surely to zero as $N$ increases because the empirical distribution converges $\mathbb{P}^\infty$-almost surely to the data-generating distribution in the second Wasserstein distance [30]. In summary, we thus have

$$\lim_{N \to \infty} \sup_{\theta \in \Theta} \left| R^\star_{r_N}(\theta, \widehat{\mathbb{P}}^f_N) - R(\theta, \mathbb{P}^f) \right| = 0 \quad \mathbb{P}^\infty\text{-a.s.} \tag{24}$$

Put differently, for $\mathbb{P}^\infty$-almost every trajectory of training samples, the functions $R^\star_{r_N}(\cdot, \widehat{\mathbb{P}}^f_N)$ converge uniformly to $R(\cdot, \mathbb{P}^f)$. The claim then follows from [55, Proposition 7.15 and Theorem 7.31]. $\qquad\square$

### 7.3 Proofs and auxiliary results for Section 5

*Proof of Theorem 5.2.* The key enabling mechanism to prove (15a) and (15b) is the so-called double smoothing method for linearly constrained convex programs [22]. Our proof parallels that of [71, Theorem 5] and is provided here to keep the paper self contained. Throughout the proof, we denote by $\mathcal{M}(\Xi)$ the vector space of all finite signed Borel measures on $\Xi$, and we equip $\mathcal{M}(\Xi)$ with the total variation norm $\|\cdot\|_{\mathsf{TV}}$. Choosing the total variation norm has the benefit that the function $g : \mathcal{P}(\Xi) \to \mathbb{R}_+$ defined through $g(\mathbb{Q}) = \mathsf{D}(\mathbb{Q}\|\widehat{\mathbb{P}}_N)$ is strongly convex with convexity parameter 1. Indeed, Pinsker's inequality implies that $d(\mathbb{Q}) \geq \frac{1}{2}\|\mathbb{Q} - \widehat{\mathbb{P}}_N\|^2_{\mathsf{TV}}$ for all $\mathbb{Q} \in \mathcal{P}(\Xi)$. To prove (15a) and (15b), we consider the primal and dual optimization problems

$$J^\star_{\mathsf{P}} = \min_{\mathbb{Q} \in \mathcal{P}(\Xi)} \left\{ \mathsf{D}(\mathbb{Q}\|\widehat{\mathbb{P}}_N) + \sup_{z \in \mathbb{R}^d} \left\{ \mathbb{E}_\mathbb{Q}[\psi(\xi)]^\top z - \sigma_E(z) \right\} \right\} \tag{25a}$$

$$J^\star_{\mathsf{D}} = \sup_{z \in \mathbb{R}^d} \left\{ -\sigma_E(z) + \min_{\mathbb{Q} \in \mathcal{P}(\Xi)} \left\{ \mathsf{D}(\mathbb{Q}\|\widehat{\mathbb{P}}_N) + \mathbb{E}_\mathbb{Q}[\psi(\xi)]^\top z \right\} \right\}, \tag{25b}$$

where $\sigma_E : \mathbb{R}^d \to \mathbb{R}$ defined through $\sigma_E(z) = \max_{x \in E} z^\top x$ denotes the support function of $E$. As the convex conjugate of the support function $\sigma_E$ is the indicator function $\delta_E : \mathbb{R}^d \to [0, \infty]$ defined through $\delta_E(x) = 0$ if $x \in E$ and $\delta_E(x) = \infty$ if $x \notin E$, the optimal value of the maximization problem over $z$ in (25a) equals $\delta_E(\mathbb{E}_\mathbb{Q}[\psi(\xi)])$. Hence, the unique minimizer of (25a) coincides with the I-projection of the empirical distribution onto the set $\Pi$. We also remark that $\sigma_E$ is continuous because $E$ is non-empty and compact [54, Corollary 13.2.2]. Assumption 5.1 then ensures via [71, Lemma 3] that there is no duality gap, i.e, $J^\star_{\mathsf{P}} = J^\star_{\mathsf{D}}$. Next, we introduce the shorthand

$$F(z) = -\sigma_E(z) + \min_{\mathbb{Q} \in \mathcal{P}(\Xi)} \left\{ \mathsf{D}(\mathbb{Q}\|\widehat{\mathbb{P}}_N) + \mathbb{E}_\mathbb{Q}[\psi(\xi)]^\top z \right\}$$

for the dual objective function. While the primal problem (25a) is an infinite-dimensional optimization problem, the dual problem (25b) can be solved via first-order methods provided that the gradient of the dual objective function $F$ can be evaluated at low cost. Unfortunately, this function fails to be smooth. Consequently, an optimal first-order method would require $O(1/\varepsilon^2)$ iterations, where $\varepsilon$ denotes the desired additive accuracy [47, Section 3.2]. However, the computation can be accelerated by smoothing the dual objective function as in [22, 46] and by exploiting structural properties. To this end, we introduce a smoothed version $F_\eta$ of the dual objective function defined through

$$F_\eta(z) = -\max_{x \in E} \left\{ x^\top z - \frac{\eta_1}{2}\|x\|^2_2 \right\} + \min_{\mathbb{Q} \in \mathcal{P}(\Xi)} \left\{ \mathsf{D}(\mathbb{Q}\|\widehat{\mathbb{P}}_N) + \mathbb{E}_\mathbb{Q}[\psi(\xi)]^\top z \right\} - \frac{\eta_2}{2}\|z\|^2_2,$$

where $\eta = (\eta_1, \eta_2) \in \mathbb{R}^2_{++}$ is a smoothing parameter. One readily verifies that $x^\star_z = \pi_E(\eta_1^{-1}z)$ solves the optimization problem in the first term. The optimization problem in the second term minimizes the sum of a relative entropy function and a linear function. Therefore, it is reminiscent of an entropy maximization problem, and one can show that it is solved by the Gibbs distribution

$$\mathbb{Q}^\star_z = \frac{\sum_{j=1}^N \exp\left(-z^\top \psi\left(\widehat{\xi}_j\right)\right) \delta_{\widehat{\xi}_j}}{\sum_{j=1}^N \exp\left(-z^\top \psi(\widehat{\xi}_j)\right)},$$

see [71, Lemma 2]. By construction, the smoothed dual objective function $F_\eta$ is $\eta_2$-strongly concave and differentiable. Its gradient can be expressed in terms of the parametric optimizers $x_z^\star$ and $\mathbb{Q}_z^\star$ as

$$\nabla F_\eta(z) = -x_z^\star + \mathbb{E}_{\mathbb{Q}_z^\star}[\psi(\xi)] - \eta_2 z = G_\eta(z),$$

where $G_\eta$ is defined in (12); see also [46, Theorem 1]. In addition, as shown in [46, Theorem 1], the gradient function $G_\eta$ is Lipschitz continuous with a Lipschitz constant $L_\eta$ that satisfies

$$L_\eta = 1/\eta_1 + \eta_2 + \left( \sup_{\lambda \in \mathbb{R}^d, \mathbb{Q} \in \mathcal{M}(\Xi)} \left\{ \lambda^\top \mathbb{E}_{\mathbb{Q}}[\psi(\xi)] \ : \ \|\lambda\|_2 = 1, \|\mathbb{Q}\|_{\mathsf{TV}} = 1 \right\} \right)^2$$

$$\leq 1/\eta_1 + \eta_2 + \left( \sup_{\lambda \in \mathbb{R}^d, \mathbb{Q} \in \mathcal{M}(\Xi)} \left\{ \|\lambda\|_2 \|\mathbb{E}_{\mathbb{Q}}[\psi(\xi)]\|_2 \ : \ \|\lambda\|_2 = 1, \|\mathbb{Q}\|_{\mathsf{TV}} = 1 \right\} \right)^2$$

$$= 1/\eta_1 + \eta_2 + (\max_{\xi \in \Xi} \|\psi(\xi)\|_\infty)^2 < \infty.$$

Therefore, the smoothed dual optimization problem

$$\sup_{z \in \mathbb{R}^d} F_\eta(z) \tag{26}$$

has a smooth and strongly concave objective function, implying that it can be solved highly efficiently via fast gradient methods. When solving (26) by Algorithm 1, we can use its outputs $z_k$ to construct candidate solutions $\widehat{\mathbb{Q}}_k$ for the primal (non-regularized) problem (25a) as described in (13). These candidate solutions satisfy the optimality and feasibility guarantees (15a) and (15b), which can be derived by using the techniques developed in [22]. A detailed derivation using our notation is also provided in [71, Appendix A]. We highlight that (15a) and (15b) critically rely on Assumption 5.1, which implies via [45, Lemma 1] that the norm of the unique maximizer of the regularized dual problem (26) is bounded above by $C/\delta$, where $C$ and $\delta$ are defined as in (14). $\qquad\square$

*Proof of Proposition 5.3.* By the definition of the DRO predictor $R^\star$ in (5), we have

$$R^\star(\theta, \mathbb{P}') = \sup_{\mathbb{Q} \in \mathcal{P}(\Xi)} \left\{ \mathbb{E}_{\mathbb{Q}}[L(\theta, \xi)] \ : \ \mathsf{D}(\mathbb{P}' \| \mathbb{Q}) \leq r, \ \mathbb{E}_{\mathbb{Q}}[\psi(\xi)] \in E \right\}$$

$$= \sup_{\mathbb{Q} \in \mathcal{P}(\Xi)} \left\{ \mathbb{E}_{\mathbb{Q}}[L(\theta, \xi)] - \sup_{z \in \mathbb{R}^d} \{ z^\top \mathbb{E}_{\mathbb{Q}}[\psi(\xi)] - \sigma_E(z) \} \ : \ \mathsf{D}(\mathbb{P}' \| \mathbb{Q}) \leq r \right\}$$

$$= \inf_{z \in \mathbb{R}^d} \sup_{\mathbb{Q} \in \mathcal{P}(\Xi)} \left\{ \mathbb{E}_{\mathbb{Q}}[L(\theta, \xi) - z^\top \psi(\xi)] + \sigma_E(z) \ : \ \mathsf{D}(\mathbb{P}' \| \mathbb{Q}) \leq r \right\}$$

$$= \inf_{z \in \mathbb{R}^d} \begin{cases} \inf_{\alpha \in \mathbb{R}} & \alpha + \sigma_E(z) - e^{-r} \exp \left( \mathbb{E}_{\mathbb{P}'}[\log(\alpha - L(\theta, \xi) + z^\top \psi(\xi))] \right) \\ \text{s.t.} & \alpha \geq \max_{\xi \in \Xi} L(\theta, \xi) - z^\top \psi(\xi) \end{cases}$$

where the second equality holds because the convex conjugate of the support function $\sigma_E$ is the indicator function $\delta_E : \mathbb{R}^d \to [0, \infty]$ defined through $\delta_E(x) = 0$ if $x \in E$ and $\delta_E(x) = \infty$ if $x \notin E$, and the third equality follows from Sion's minimax theorem, which applies because the relative entropy $\mathsf{D}(\mathbb{P}' \| \mathbb{Q})$ is convex in $\mathbb{Q}$, while the distribution family $\mathcal{P}(\Xi)$ is convex and weakly compact. Finally, the fourth equality follows from [77, Proposition 5], which applies because $r > 0$ and because the modified loss function $L(\theta, \xi) - z^\top \psi(\xi)$ is continuous in $\xi$ for any fixed $\theta$ and $z$. The last expression is manifestly equivalent to (16), and thus the claim follows. $\qquad\square$

The following corollary of Proposition 5.3 establishes that the DRO predictor $R^\star$ is continuous. This result is relevant for Theorem 4.1.

**Corollary 7.1** (Continuity of $R^\star$)**.** *If $r > 0$, $0 \in \mathrm{int}(E)$ and for every $z \in \mathbb{R}^d$ there exists $\xi \in \Xi$ such that $z^\top \psi(\xi) > 0$, then the DRO predictor $R^\star$ is continuous on $\Theta \times \Pi$.*

*Proof.* Since $r > 0$, we may use Proposition 5.3 to express the DRO predictor as

$$R^\star(\theta, \mathbb{P}') = \inf_{z \in \mathbb{R}^d} \varphi_E(\theta, z, \mathbb{P}') \tag{27}$$

for all $\theta \in \Theta$ and $\mathbb{P}' \in \Pi$, where the parametric objective function $\varphi_E$ is defined through

$$\varphi_E(\theta, z, \mathbb{P}') = \inf_{\alpha \geq \underline{\alpha}(\theta, z)} \alpha + \sigma_E(z) - e^{-r} \exp\left(\mathbb{E}_{\mathbb{P}'}[\log(\alpha - L(\theta, \xi) + z^\top \psi(\xi))]\right)$$

with $\underline{\alpha}(\theta, z) = \max_{\xi \in \Xi} L(\theta, \xi) - z^\top \psi(\xi)$. Note that the support function $\sigma_E$ is continuous because $E$ is compact. Applying [77, Proposition 6] to the modified loss function $L(\theta, \xi) - z^\top \psi(\xi)$ thus implies that $\varphi_E$ is continuous on $\Theta \times \mathbb{R}^d \times \Pi$. To bound $\varphi_E$ from below by a coercive function, we define

$$\kappa = \min_{\|z\|_2 = 1} \min_{\mathbb{Q} \in \Pi} \sigma_E(z) - e^{-r} z^\top \mathbb{E}_{\mathbb{Q}}[\psi(\xi)],$$

which is a finite constant. Indeed, $\sigma_E$ is continuous because $E$ is compact, and $\mathbb{E}_{\mathbb{P}'}[\psi(\xi)]$ is weakly continuous in $\mathbb{P}'$ because $\psi$ is a continuous and bounded function on the compact set $\Xi$. In addition, the unit sphere in $\mathbb{R}^d$ is compact, and the set $\Pi$ is weakly compact. Therefore, both minima in the definition of $\kappa$ are attained at some $z^\star \in \mathbb{R}^d$ with $\|z^\star\|_2 = 1$ and some $\mathbb{Q}^\star \in \Pi$, respectively. As $0 \in \text{int}(E)$ and $z^\star \neq 0$, we have $\sigma_E(z^\star) > 0$. In addition, as $\mathbb{Q}^\star \in \Pi$, we have $\mathbb{E}_{\mathbb{Q}^\star}[\psi(\xi)] \in E$, which implies that $(z^\star)^\top \mathbb{E}_{\mathbb{Q}^\star}[\psi(\xi)] \leq \sigma_E(z^\star)$. Again as $r > 0$, this reasoning ensures that

$$\kappa = \sigma_E(z^\star) - e^{-r}(z^\star)^\top \mathbb{E}_{\mathbb{Q}^\star}[\psi(\xi)] > 0.$$

Similarly, we introduce the finite constant

$$\underline{L} = \min_{\theta \in \Theta} \min_{z \in \mathbb{R}^d} \min_{\xi \in \Xi} (1 - e^{-r}) \underline{\alpha}(\theta, z) + e^{-r} L(\theta, \xi).$$

To see that $\underline{L}$ is bounded below, note that the definition of $\underline{\alpha}$ and the subadditivity of the minimum operator lead to the estimate

$$\underline{L} \geq (1 - e^{-r}) \min_{\theta \in \Theta} \min_{\xi \in \Xi} L(\theta, \xi) + (1 - e^{-r}) \min_{z \in \mathbb{R}^d} \max_{\xi \in \Xi} (-z)^\top \psi(\xi) + e^{-r} \min_{\theta \in \Theta} \min_{\xi \in \Xi} L(\theta, \xi)$$

$$= \min_{\theta \in \Theta} \min_{\xi \in \Xi} L(\theta, \xi) + (1 - e^{-r}) \min_{z \in \mathbb{R}^d} \max_{\xi \in \Xi} (-z)^\top \psi(\xi).$$

The first term in the resulting lower bound is finite because $L$ is continuous, while $\Theta$ and $\Xi$ are compact. The second term is also finite because the convex function $\max_{\xi \in \Xi} (-z)^\top \psi(\xi)$ is continuous in $z$ thanks to the continuity of $\psi$ and the compactness of $\Xi$. In addition, $\max_{\xi \in \Xi} (-z)^\top \psi(\xi)$ is also coercive in $z$ because of the assumption that for every $z \in \mathbb{R}^d$ there exists $\xi \in \Xi$ with $z^\top \psi(\xi) > 0$.

The above preparatory arguments imply that

$$\varphi_E(\theta, z, \mathbb{P}') \geq \inf_{\alpha \geq \underline{\alpha}(\theta, z)} (1 - e^{-r}) \alpha + \sigma_E(z) + e^{-r} \mathbb{E}_{\mathbb{P}'}[L(\theta, \xi)] - e^{-r} z^\top \mathbb{E}_{\mathbb{P}'}[\psi(\xi)]$$

$$= (1 - e^{-r}) \underline{\alpha}(\theta, z) + e^{-r} \mathbb{E}_{\mathbb{P}'}[L(\theta, \xi)] + \left( \sigma_E\left(\frac{z}{\|z\|_2}\right) - e^{-r} \left(\frac{z}{\|z\|_2}\right)^\top \mathbb{E}_{\mathbb{P}'}[\psi(\xi)] \right) \|z\|_2$$

$$\geq \underline{L} + \kappa \|z\|_2,$$

where the first inequality exploits Jensen's inequality, the equality holds thanks to the positive homogeneity of the support function $\sigma_E$ and the trivial observation that $e^{-r} < 1$, and the second inequality follows from the definitions of $\underline{L}$ and $\kappa$ and the assumption that $\mathbb{P}' \in \Pi$. We thus have

$$\varphi_E(\theta, z, \mathbb{P}') \geq \underline{L} + \kappa \|z\|_2 \quad \forall \theta \in \Theta, \ \forall z \in \mathbb{R}^d, \ \forall \mathbb{P}' \in \Pi. \tag{28a}$$

Next, define

$$\overline{L} = \max_{\theta \in \Theta} \max_{\xi \in \Xi} L(\theta, \xi),$$

and note that

$$\inf_{z \in \mathbb{R}^d} \varphi_E(\theta, z, \mathbb{P}') = R^\star(\theta, \mathbb{P}^\star) \leq \overline{L} \quad \forall \theta \in \Theta, \ \forall \mathbb{P}' \in \Pi. \tag{28b}$$

Taken together, the estimates (28a) and (28b) imply that

$$R^\star(\theta, \mathbb{P}') = \inf_{z \in \mathbb{R}^d} \left\{ \varphi_E(\theta, z, \mathbb{P}') : \|z\|_2 \leq \frac{\overline{L} - \underline{L}}{\kappa} \right\},$$

which in turn implies via Berge's maximum theorem [7, pp. 115–116] and the continuity of the objective function $\varphi_E$ on $\Theta \times \mathbb{R}^d \times \Pi$ that the DRO predictor $R^\star$ is indeed continuous on $\Theta \times \Pi$. $\quad\square$

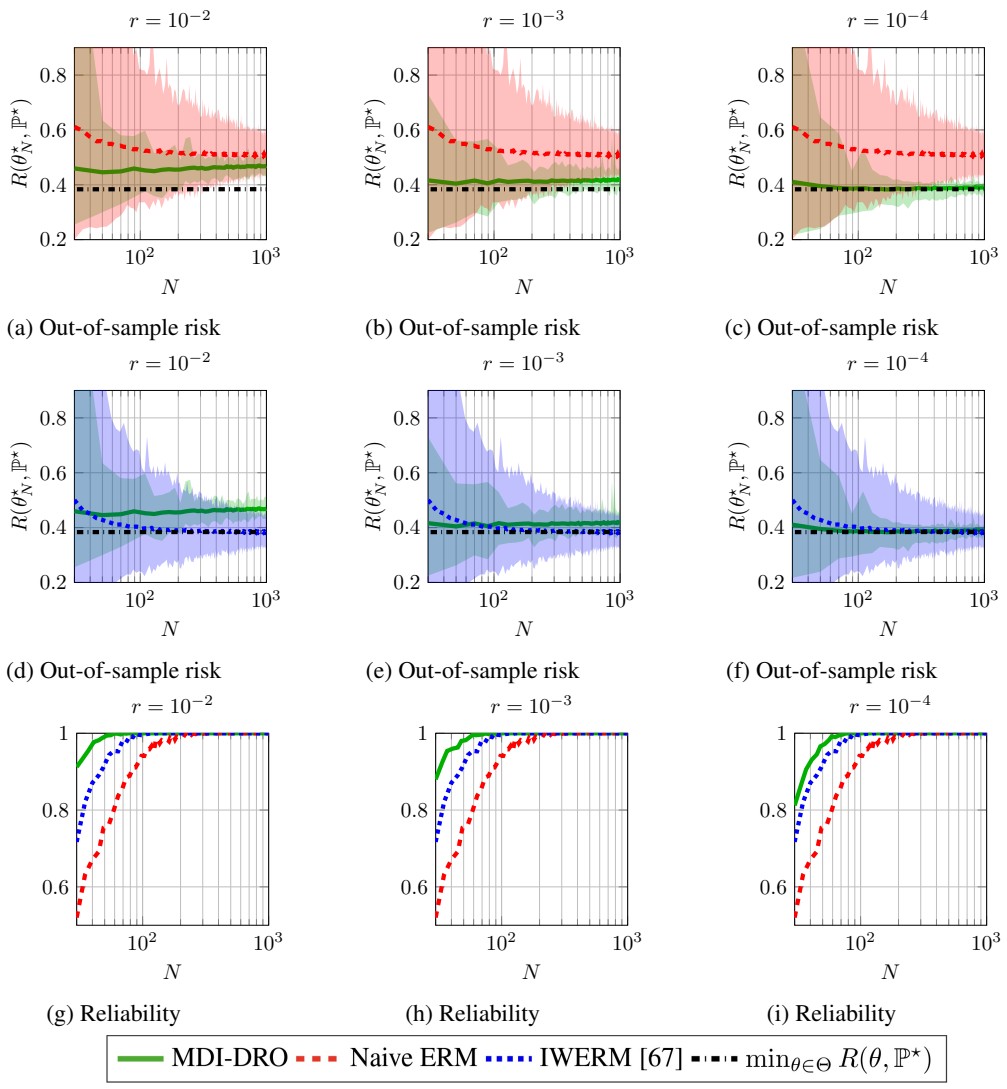

Figure 4: Additional results for the synthetic dataset with $m = 6$ (see also Figure 1). Shaded areas and lines represent ranges and mean values across 1000 independent experiments, respectively.

## 7.4 Auxiliary results for Section 6

**Classification under covariate shift.** We construct a synthetic training data consisting of feature vectors $\widehat{x}_i$ and corresponding labels $\widehat{y}_i$. Under the training distribution $\mathbb{P}$, the feature vectors are uniformly distributed on $[0, 1]^{m-1}$, where $m \geq 2$, and the labels are set to $\widehat{y}_i = 1$ if $\frac{1}{m-1} \sum_{j=1}^{m-1} (\widehat{x}_i)_j > \frac{1}{2}$ and $\widehat{y}_i = -1$ otherwise. By construction, we thus have $\mathbb{E}_{\mathbb{P}}[(x, y)] = (0, 0)$. The test distribution $\mathbb{P}^\star$ differs from $\mathbb{P}$. Specifically the probability density function of the features under $\mathbb{P}^\star$ is set to

$$p^\star(x) = \frac{2}{m-1} \sum_{j=1}^{m-1} x_j \quad \forall x \in [0, 1]^{m-1},$$

while the conditional distribution of the labels given the features is the same under $\mathbb{P}$ and $\mathbb{P}^\star$. A direct calculation then reveals that $\mathbb{E}_{\mathbb{P}^\star}[x_j] = \frac{m-2}{2(m-1)} + \frac{2}{3(m-1)} = \mu^\star > 0$ for all $j = 1, \ldots, m-1$. Similarly, one can show that $\mathbb{E}_{\mathbb{Q}}[y] > 0$. In the numerical experiments we assume that both $\mathbb{P}$ and $\mathbb{P}^\star$ are unknown. However, we assume to have access to $N$ i.i.d. samples from $\mathbb{P}$, and we assume that $\mathbb{P}^\star$ is known to satisfy $\mathbb{E}_{\mathbb{P}^\star}[\psi(\xi)] \in E$, where $\psi(x, y) = (x, y)$ and $E = [(\mu^\star - \varepsilon) \cdot \mathbf{1}, (\mu^\star - \varepsilon) \cdot \mathbf{1}]$ for some $\varepsilon > 0$ that is sufficiently small to ensure that $0 \notin E$. This implies that $\mathbb{P} \notin \Pi$.

**Inventory control model.** Consider an inventory that stores a homogeneous good, and let the state variable $s_i$ represent the stock level at the beginning of period $i$. The control action $a_i$ reflects the

order quantity in period $i$, and we assume that any orders are delivered immediately at the beginning of the respective periods. The disturbance $\zeta_i$ represents an uncertain demand revealed in period $i$. We assume that the demands are i.i.d. across periods and follow a geometric distribution on $\mathbb{N} \cup \{0\}$ with success probability $\lambda \in (0, 1)$. The inventory capacity is denoted by $\gamma \in \mathbb{N}$, and any orders that cannot be stored are lost. Similarly, we assume that any demand that cannot be satisfied is also lost. The system equation describing the dynamics of the stock level is thus given by

$$s_{i+1} = \max\{0, \min\{\gamma, s_i + a_i\} - \zeta_i\} \quad \forall i = 0, 1, 2, \ldots,$$

see also [28]. Our aim is to estimate the long-run average cost generated by a prescribed ordering policy, assuming that the (uncertain) cost incurred in period $i \in \mathbb{N}$ can be expressed as

$$r(s_i, a_i, \zeta_i) = pa_i + h(s_i + a_i) - v\min\{s_i + a_i, \zeta_i\}.$$

The three terms in the above expression capture the order cost, the inventory holding cost and the profit from sales, where $p > 0$ and $h > 0$ denote the costs for ordering or storing one unit of the good, while $v > 0$ denotes the unit sales price. The expected per period cost thus amounts to

$$c(s_i, a_i) = pa_i + h(s_i + a_i) - v\frac{(1-\lambda)}{\lambda}\left(1 - (1 - \lambda)^{(a_i + s_i)}\right).$$

The simulation results shown in Figure 3 are based on an instance of the inventory control model with state space $\mathcal{S} = \{1, 2, \ldots, 5\}$, action space $\mathcal{A} = \{1, 2, \ldots, 4\}$, and parameters $\lambda = 0.2$, $\gamma = 5$, $p = 0.6$, $h = 0.3$ and $v = 1$. The threshold for computing the modified IPS estimator is set to $\beta = 4$. It is easy to verify that, under this model parameterization, the cost function $c(s_i, a_i)$ is invertible in the sense that $s_i$ and $a_i$ are uniquely determined by $c(s_i, a_i)$; see also Example 3.6.