# OpenReview forum: "Robust Generalization despite Distribution Shift via Minimum Discriminating Information"
_NeurIPS.cc/2021/Conference — NeurIPS 2021 Poster_

### Official Review · Reviewer_qvWC · 2021-07-04

**Rating:** 7
**Confidence:** 3

**Summary:**

The paper proposes to solve a distribution shift problem by minimizing risk over an uncertainty ball around the empirical training distribution, subject to a shift with a known structure. This structure is encoded as knowing the expectation of some function of the random observation falls within a compact convex set E. The paper uses the principle of minimum discriminating to justify encoding *only* these structural requirements, being the least prejudiced approximation to the original distribution given only the structural knowledge.

The authors gives statistical guarantees and a tractable algorithm for finding the I-projection (the closest distribution in relative entropy which satisfies the structural requirement) of the empirical training distribution and then finding an approximate DRO solution around this projection.

**Limitations And Societal Impact:**

Sure

**Main Review:**

I think this is a nice paper with a well-defined focus and useful results. My background is not in this area so I did not check the math in detail but I am in favor of acceptance. A couple notes:

* The paper is quite dense, particularly the examples. At a minimum, it would be helpful to indicate earlier on which of the examples will be persistent, so that the reader can focus on understanding these and perhaps not so much on the ones which will not appear again. This would also help justify the lengthy descriptions and encourage the reader not to skip them. I'd recommend even dropping all but the OPE example, or at least drastically shortening them.
* I'm struggling to understand the equation between lines 133 and 134. We're assuming N samples (an empirical expectation) but then also evaluating the true expectation? I've considered interpreting the LHS as $\mathbb{E}_{\mathbb{P}}[L(\theta, \xi) \mid \psi(\xi) \in E]$ but I think this is not correct. What is this meant to say?
* Line 137, $\ll$ is typically "absolutely continuous w.r.t." but I don't think it's defined anywhere. I think this term is probably worth explicitly defining.

**Time Spent Reviewing:**

4.5

---

> ### Author Response · Authors · 2021-08-09
> **Reply to Reviewer qvWC**
>
> We thank the referee for the constructive feedback and the time spent in reviewing our paper.
>
> *Density of exposition.*
> We will move one example to the appendix and would use the additional page (in the camera ready version) to make the exposition less dense. This will also contribute towards increasing readability and clarity.
>
> *Conditional limit theorem.*
> One should see the statement in Proposition 3.4 as
>     $\lim_{N\to\infty} E_{\mathbb{P}}$ $[L(\theta,\xi_1)|\frac{1}{N}\sum_{i=1}^N\psi(\xi_i)\in E]$ $= E_{\mathbb{P}^f}[L(\theta,\xi)]\quad \forall \theta\in\Theta,$
> where, importantly, the expectation on the left is with respect to $\mathbb{P}$ and $\xi_1, \dots, \xi_N$ are i.i.d. random variables according to $\mathbb{P}$.
>
> The interpretation provided by the reviewer is not valid as the limit cannot be taken inside the conditional part of the expectation. To see this, suppose for the sake of contradiction that the limit can be taken inside the expectation. Then, by the strong law of large number, which states that $\lim_{N\to\infty} \frac{1}{N}$ $\sum_{i=1}^N$ $\psi(\xi_i) = E_\mathbb{P}[\psi(\xi_1)]$ almost surely, we get
>
>   $ \lim_{N\to\infty} E_{\mathbb{P}}[L(\theta,\xi_1)|\textstyle{\frac{1}{N}\sum_{i=1}^N\psi(\xi_i)\in E]} =  E_\mathbb{P}[L(\theta,\xi_1)|E_\mathbb{P}[\psi(\xi_1)]\in E].$
>
> Now, since $E_\mathbb{P}[\psi(\xi_1)]$ is deterministic, there are two cases: (i) $E_\mathbb{P}[\psi(\xi_1)]\in E$, then $E_\mathbb{P}[L(\theta,\xi_1)|E_\mathbb{P}[\psi(\xi_1)]\in E]$ $=E_\mathbb{P}[L(\theta,\xi_1)]\neq E_{\mathbb{P}^f}[L(\theta,\xi)]$, which is a contradiction to the conditional limit theorem (Proposition 3.4). In the other case (ii), where $E_\mathbb{P}[\psi(\xi_1)]\notin E$ we are conditioning on a negligible set in which case that object is not defined. As a conclusion our assumption is wrong and we can indeed not take the limit inside the conditioning of the expectation.
>
>
> *Absolute continuity.* We thank the reviewer for pointing out that the symbol for absolute continuity of probability measures has not been introduced. We will do that.

---

> > ### Comment · Reviewer_qvWC · 2021-08-12
> > **Notation is still somewhat unclear to me**
> >
> > In your response, you've given a subscript to the $\xi$ on the LHS of the expectation, whereas it does not have one in the paper. Which is correct?
> >
> > My confusion here is that I don't see how the expectation you've written makes sense. The LHS of the expectation is independent of $N$---it is simply the expectation of a term with respect to $\mathbb{P}$. What are the $N$ samples you're conditioning on? You've indicated that they are $N$ samples $\xi_1,\ldots,\xi_N\sim\mathbb{P}$. But I think maybe these should be a *different* expectation term than the LHS.
> >
> > Let's write this out formally, using the fact that $E[x | A] = \frac{E[x\ \mathbf{1}(A) ]}{P(A)}$ (here $\mathbf{1}(A)$ is an indicator function for $A$ occurring):
> >
> > $$E_{\mathbb{P}}[L(\theta,\xi)| \frac{1}{N}\sum_{i=1}^N \psi(\xi_i) \in E] = \frac{E_{\mathbb{P}}\left[L(\theta,\xi)\ \mathbf{1}\left(\frac{1}{N}\sum_{i=1}^N \psi(\xi_i) \in E\right)\right]}{\mathbb{P}\left(\frac{1}{N}\sum_{i=1}^N \psi(\xi_i) \in E\right)} $$
> >
> > The denominator is perfectly reasonable. Let's rewrite the numerator in measure-theoretic terms:
> >
> > $$E_{\mathbb{P}}\left[L(\theta,\xi)\ \mathbf{1}\left(\frac{1}{N}\sum_{i=1}^N \psi(\xi_i) \in E\right)\right] = \int_{\textrm{Supp}(\mathbb{P})} L(\theta, \xi) \times \mathbf{1}\left(\frac{1}{N}\sum_{i=1}^N \psi(\xi_i) \in E\right)\  d p(\xi).$$
> >
> > Now, my confusion is: *where are the $N$ samples in the indicator function coming from?* Sure, they are iid samples from $\mathbb{P}$, but as far as I can tell they are unrelated to the integral we're taking---one is a genuine expectation (Lebesgue integral), and the other is an empirical expectation. And if they're drawn independently, then conditioning on the event shouldn't affect the expectation.

---

> > > ### Author Response · Authors · 2021-08-13
> > > **Clarification of notation and conditional limit theorem**
> > >
> > > We thank you for your comment and apologize for the confusion. The short answer to your question is that $\xi$ in the original version of Proposition 3.4 should have been $\xi_1$. We corrected this typo in the first response above. Regarding your second question, you are absolutely right that the expectation in Proposition 3.4 is evaluated with respect to the $N$-fold product $\mathbb P^N$. We apologize for the sloppy shorthand notation. We will consistently replace $\mathbb{P}$ with $\mathbb{P}^N$ in the revision.
> > >
> > > To provide a better understanding of Proposition 3.4, we would like to point out that the conditional limit theorem (see [15, Theorem 11.6.2] for an simple version with $\psi(\xi_i)=\xi_i$ and [17] for a more general version) basically states that if we are given $N$ i.i.d. random variables $\xi_1, ..., \xi_N$ governed by $\mathbb{P}$, then (in the limit as $N\to \infty$) the random variables $\xi_j$ conditioned on $\frac{1}{N}\sum_{i=1}^N\psi(\xi_i)\in E$ are i.i.d. and follow the distribution $\mathbb{P}^f$ defined in the paper. Due to the permutation symmetry, we may use $\xi_1$ in Proposition 3.4.
> > >
> > > In summary, we will change the notation in Proposition 3.4 to
> > > $\lim_{N\to\infty} E_{\mathbb{P}^N}[L(\theta,\xi_1)|\textstyle{\frac{1}{N}\sum_{i=1}^N\psi(\xi_i)\in E]}
> > >     =  \mathbb{E}_{\mathbb{P}^f}[L(\theta,\xi_1)] \quad \forall \theta\in\Theta.$
> > >
> > > Using this more explicit notation, the last equation (nominator) in your query can be expressed as
> > > $E_{\mathbb{P}^N}\left[L(\theta,\xi_1) \mathsf{1}\left(\frac{1}{N}\sum_{i=1}^N\psi(\xi_i)\in E\right)\right] \\
> > >     = \int_{\text{Supp}(\mathbb{P})} \dots \int_{\text{Supp}(\mathbb{P})}L(\theta,\xi_1) \times \mathsf{1} \left(\frac{1}{N}\sum_{i=1}^N\psi(\xi_i)\in E\right) dp(\xi_1)\dots dp(\xi_N),$
> > > and therefore the $N$ samples all affect the expectation. Please let us know if we can provide any further information.

---

### Official Review · Reviewer_fqJW · 2021-07-12

**Rating:** 5
**Confidence:** 4

**Summary:**

In this paper, they aim at test and training distribution shift problem where partial knowledge of shifted test distribution is known. With the principle of minimum discriminating information, they introduce a new modeling framework catching prior structural information, and a specific MDI-DRO with theoretical generalization bound, satisfying the statistical guarantee of out-of-sample.  For computational tractability, they leverage convex duality and accelerated first-order methods to efficiently solve the DRO program. Experiments on two distinct problem classes prove the availability of the proposed approach.

========
After reading the authors' response and other reviews, I would like to raise my score to 5, but still think the paper in its current form is not self-contained due to the lack of ablation study and important baselines.

**Limitations And Societal Impact:**

I wonder if their approach can be applied to large-scale complex data structures, for example, image data that has been fully studied for the OOD problems.

**Main Review:**

1. This work assumes and takes advantage of prior structure knowledge of tests in their modeling framework. So in their experiments, approaches that do not employ such information should be compared to verify the validity of leveraging partial test information. For example, there are a series of works facing an arbitrary and unknown distribution shift based on training data (Stable Prediction across Unknown Environments. KDD, 2018).
2. The baselines and experimental settings are inadequate. Other DRO algorithms and other types of methods that use or do not use the test information should be discussed and compared, for example, self-normalized IPS (reweighing approach) and double robust methods for inventory control example. And the OOD generalization problem has developed to cover complicated data structures. More real data should be applied.
3. The presentation of theory and method is confusing. Some propositions and examples can be moved to the appendix and they can highlight the theorem and related conclusion in writing.


**Time Spent Reviewing:**

8 hours

---

> ### Author Response · Authors · 2021-08-09
> **Reply to Reviewer fqJW**
>
> We thank the referee for the constructive feedback and the time spent in reviewing our paper.
>
> 1. We thank the reviewer for making us aware of the stream of research around (Stable Prediction across Unknown Environments, 2018). Indeed the problems studied are related to ours. We will add a detailed discussion in the revised manuscript.
> Nevertheless, we would like to point out, that our approach is fundamentally different in various aspects: First, we assume to have some prior knowledge of the test distribution and then investigate how to optimally exploit this prior knowledge in the underlying classification (or decision) problem. The word "optimally" is with respect to an information theoretic notion relating our method to the principle of minimum discriminating information via the conditional limit theorem (Proposition 3.4).
> Second, by invoking this prior knowledge and the concept of an information projection, we are able to establish generalization bounds for the underlying learning problem under the test-distribution based on the data coming only from the training distribution, see Theorem 4.1 and Corollary 4.3.
> Third, we show that our approach is statistically optimal in the sense of being Pareto-optimal as described in Theorem 4.2. This, to the best of our knowledge is the first such result.
> Moreover, when comparing our method with the mentioned reference (Stable Prediction across Unknown Environments, 2018), our method, while requiring prior information on the test distribution does not require Assumption 1 in (Stable Prediction across Unknown Environments, 2018).
>
>
>  2. We are happy to add more baselines to the numerical examples (e.g., other DRO algorithms (Wasserstein, moment-based DRO), double robust method for inventory control). We are currently running these experiments and will upload the corresponding plots to the anonymous github repository of this paper (see footnote in the paper). First experiments indicate that the classical double robust method for the infinite horizon inventory model we look at also suffers from higher variance compared to our proposed method. We are also willing to add a second real-world data experiment.
> Regarding DRO methods that do not use the test information (while having worked in the DRO field for many years), we are not aware of any such method. Regarding the OOD generalization methods addressing more complicated data structures, we would like to emphasize that we focus on "simple" i.i.d. data here, but this allows us to prove powerful generalization bounds under distribution shifts see Theorem 4.1 and Corollary 4.3.
> We would also like to point out that in the OPE example, the benchmark we compared against (called IPS in our legend) is actually the method referred to in the literature as ``Off-Policy Estimation via Stationary State Density Ratio Estimation" (see Breaking the Curse of Horizon: Infinite-Horizon Off-Policy Estimation, 2018).
>
> 3. We will move one example to the appendix and use the additional page to make the exposition less dense, which will hopefully contribute towards increasing readability and clarity.

---

> > ### Comment · Reviewer_fqJW · 2021-08-17
> > **about the necessrity and effect of the prior structure**
> >
> > I think this work is self-consistent if the prior structure is proved to be imperative despite the distribution shift. I raised this major concern in the review, but the response didn't clearly address this point. Actually, there are a variety of works that do not use the test information, like the works I mentioned. In my opinion, the necessity and effect of prior structure should be validated, at least in an empirical way by designing sufficient ablation studies and comparisons.
> > Considering the theoretical analysis that may be helpful for understanding the targeting problem, I would like to raise my score to 5.

---

> > > ### Author Response · Authors · 2021-08-18
> > > **Response on ``necessity and effect of prior structure"**
> > >
> > > Thank you for your comment and your appreciation of our theoretical analysis. The method proposed in our paper uses the available prior information on the (unknown) test distribution in an information-theoretically optimal manner. Therefore, we expect it to generalize better than methods that ignore this information [e.g., the method described in "Stable Prediction across Unknown Environments," KDD, 2018, which remains applicable in more general settings]. We have already tested our approach against additional benchmarks for the OPE problem as per your suggestion. We have included these results in the anonymized Github reporsitory referenced in the paper. We are currently working on the requested comparison against [Stable Prediction across Unknown Environments. KDD, 2018] and will add it to the Github repository as soon as the results are available. We will also add a detailed discussion of this stream of research in the revised manuscript. Please let us know if we can provide any further information.

---

### Official Review · Reviewer_h1zT · 2021-07-19

**Rating:** 6
**Confidence:** 3

**Summary:**

The authors propose a new MDI-DRO approach that is the combination of an I-projection (to adjust for distribution shift) and DRO over both a KL-divergence constraint and a moment constraint. The major contribution is the generalization / efficiency theory results, and a convex optimization procedure for solving the DRO problem reasonably efficiently.

**Limitations And Societal Impact:**

No issues expected.

**Main Review:**

The introduction was mostly clear, but the discussion of P^f around line 97 stuck out, because there's is an abrupt (and very important) statement 'we identify P^f with an I-projection of P onto Pi' . But this is actually a really tricky and somewhat subtle point. You can either view this as an assumption on P^f, where you get to pick an arbitrary phi, or a statement that phi is fixed (as in prop 3.5) and P^f is arbitrary. In either case, this seems like in many ways the key assumption and structure, so I'd be alot more careful about the writing here.

As a related work note, I believe 'Robust Classification Under Sample Selection Bias' and the line of work by Brian Ziebart may be relevant. There's some similarity to this work in that they consider minimax games with moment constraints on the distribution. Their setup is similar to the DRO problem in eq (5), but they don't have a I-projection step as in eq (4). I think it would add to the paper to carefully discuss the differences between these approaches and MDI-DRO.

The writing in section 5 can also be made much clearer. I wasn't sure exactly what 'z' was until Thm 5.2, where it became clear that it was going to be used to construct \mu_{k,\eta} which was the solution. I'd suggest a slight rewrite that brings the high level plan before the details in the 'computation of i projection' section.

I also think the slater point discussion could be improved a bit - my understanding is that this is a fairly mild assumption, equivalent to the existence of an interior point, but the writing doesnt make it clear if this is something that's a regularity condition that can safely be treated as a detail, or something that has substantial implications for the applicability of this optimization approach.

Experiment wise, the second, 'real world data' experiment has several points that could use further discussion. The left panel shows out of sample cost that increases substantially as r goes to zero. This seems fairly counterintuitive, esp given the discussion in the earlier experiment that there is a 'known tradeoff stating that small regularization parameter r leads to small out of sample risk'. It's also pretty surprising to me that the model doesn't degenerate as r grows large. at r=10.0, I'm guessing that the KL constraint is sufficiently loose that the KL constraint does very little, and almost all of the gains come from the moment constraint. Is this true?

As a framing thing, I think it may also be helpful to mention that the goal of the paper is in many ways the upper bound on the risk, rather than the out of distribution risk itself. This is because it seems likely that if your goal is only out of distribution performance, then you might as well take r=0, which would just be the I-projection.

Minor notes — I think the arguments in line 127 about the conditional limit theorem don't necessarily justify min. discriminating information (which is an argument made in line 129). My understanding was that the this is a type of consistency result (as n→infty, I-projections converge to the population) and I fail to see how this alone provides an intuitive justification for modeling distribution shifts via I-projections.

As a style note- the large red boxes around the legends make it harder to read the figures and the legend.

**Time Spent Reviewing:**

2

---

> ### Author Response · Authors · 2021-08-09
> **Reply to Reviewer h1zT**
>
> We thank the referee for the constructive feedback and the time spent in reviewing our paper.
>
> *Role of $\mathbb{P}^f$ in (1).* We agree with the reviewer that when we fix $\mathbb{P}^f$ in (1), there are dichotomous interpretations for that statement.
> In the first setting (which is used in the off-policy evaluation problem, Example 3.6) the distribution $\mathbb{P}^f$ is given and the set $\Pi$ is a design choice. Thanks to Proposition 3.6, we know how to select $\Pi$ such that $\mathbb{P}^f$ is indeed the I-projection of $\mathbb{P}$ onto $\Pi$. Alternatively, we consider the setting where $\Pi$ is given by prior information on the test distribution (see Examples 3.2, 3.3), and we use $\mathbb{P}^f$ as an approximation (motivated by the principle of minimum discriminating information) for the unknown test distribution. We will clarify this role early on.
>
> *``Robust Classification Under Sample Selection Bias" by Ziebart.*
> We thank the reviewer for this relevant reference - Reviewer 1 also pointed us to that same stream of research. Indeed, this approach is related to our paper, as it proposes a minimax approach for regression problems under covariate shift. The considered DRO framework optimizes over so-called moment-based ambiguity sets. Our approach is fundamentally different in two perspectives: First, the framework is not specifically tailored to covariate shifts, these are rather one possible instance for our framework. Second, our choice of ambiguity set is statistically optimal due to the Pareto-optimality result, Theorem 4.2.
>
> *Role of $z$ in the section ``Computation of I-projection".*
> In order to compute the I-projection, we need to solve the optimization problem (3). We use convex duality to reformulate the problem and then solve the so-called dual counterpart. The variable $z$ denotes the dual variable. We will clarify this when using $z$ for the first time in the revised manuscript.
>
> *Discussion about Slater point.*
> We agree with the reviewer that in many practical applications the existence of a Slater point is indeed given. We would like to point out, however, that in Assumption 5.1 we need to know (or have an estimate) of the parameter $\delta$, which then defines the number of iterations of Algorithm 1 required to ensure an $\varepsilon$-optimal solution (see Theorem 5.2).
>
> *Question on experiment.*
> In Figure 2(a), when the radius $r$ goes to zero, the observed behaviour that the corresponding out-of-sample cost increases is reminiscent of overfitting. It is well known, that the DRO parameter $r$ acts as regularizer to the problem. On the other hand, if we select $r$ large (i.e., $r=10$), the resulting out-of-sample cost is similar than for $r=0.1$. For this behaviour, the reviewer's intuition is right that the KL constraint eventually becomes redundant (we numerically verified this statement).
>
> *Upper bound.* The issue with setting $r=0$ (in which case the DRO method collapses to classical SAA) is that the resulting data-driven decision in general suffers from overfitting. This phenomena can indeed be observed in Figure 2(a). Selecting $r>0$ induces a regularization effect which provably avoids overfitting (Theorem 4.1).
>
> *Justification of PMDI via conditional limit theorem.*
> We agree with the reviewer that the conditional limit theorem is an asymptotic statement and that it is probably better to see it as a consistency result for the PMDI.

---

### Official Review · Reviewer_chLA · 2021-07-23

**Rating:** 7
**Confidence:** 3

**Summary:**

This paper proposes a distributionally robust optimization (DRO) framework for learning models that can generalize well under distribution shifts, when partial knowledge about the nature of the shift is available. As opposed to classical domain adaptation methods, the proposed method does not require data from the target distribution. For an uncertainty set of distributions defined to satisfy a set of moment conditions (the partial knowledge about the shift), the authors propose to train a model to minimize risk with respect to the distribution in the uncertainty set which minimizes the relative entropy from the training distribution. Using the empirical distribution of the training data as the source distribution, the authors show that the solution to this minimax problem upper bounds the optimal risk and provide finite sample guarantees. The authors then develop a two step learning procedure: First, they estimate the I-projection (using a fast gradient-like method, Algorithm 1). Then, they can efficiently compute the model parameters. They also consider simple extensions to the case of off policy evaluation in reinforcement learning. Synthetic and real data experiments demonstrate the effectiveness of the proposed method given a good choice of the relative entropy radius hyperparameter.

**Limitations And Societal Impact:**

Yes, though I wonder if there might be any implications for performance on subgroups created by the choice of shift constraints.

**Main Review:**

The distribution shift problem addressed in this paper is very relevant in the community, and a distinct advantage of DRO methods (like the proposed method) is that they do not require samples from the test distribution. The structure for expressing knowledge about the distribution shift seems useful and interpretable. I thought the paper was well-written and I really appreciated the examples---they helped me understand how the proposed framework would work on practical examples. The experiments were presented in a way in which it was easy to draw conclusions about how the method performs as sample size and hyperparameters vary.

I had a couple questions/concerns that I hope the authors can address:
- Can the authors provide any guidance on how to choose the $r$ hyperparameter (defining the radius of the relative entropy ball)? From a practical perspective, this seems like one of the primary challenges that would make the method hard to use. As the experiments demonstrate, the proposed method can achieve very good performance, but this is highly sensitive to the choice of $r$.
- The authors claim that, e.g., the principle of maximum entropy has not been used to model general distribution shifts. I would like to point the authors to Chen et al. (2018), which considers learning a regression model under covariate shift with moment constraints. Their setting is a little different from the current paper (the current paper considers a more general problem), but I think it deserves to be mentioned.

Chen, X., Monfort, M., Liu, A., & Ziebart, B. D. (2016, May). Robust covariate shift regression. In Artificial Intelligence and Statistics (pp. 1270-1279). PMLR.

As a suggestion, I think that the discussion of related work could be broadened to include work at the intersection of distribution shift, causality, and distributional robustness. Tools from causality have been used to represent structural knowledge about distribution shifts, and recent works have established connections between causal predictors and distributional robustness under shifts arising from interventions.

Causal inference & distributional robustness:
- Rothenhäusler, D., Meinshausen, N., Bühlmann, P., & Peters, J. (2021). Anchor regression: Heterogeneous data meet causality. Journal of the Royal Statistical Society: Series B (Statistical Methodology), 83(2), 215-246.
- Meinshausen, N. (2018, June). Causality from a distributional robustness point of view. In 2018 IEEE Data Science Workshop (DSW) (pp. 6-10). IEEE.
- Rojas-Carulla, M., Schölkopf, B., Turner, R., & Peters, J. (2018). Invariant models for causal transfer learning. The Journal of Machine Learning Research, 19(1), 1309-1342.
- Subbaswamy, A., Schulam, P., & Saria, S. (2019, April). Preventing failures due to dataset shift: Learning predictive models that transport. In The 22nd International Conference on Artificial Intelligence and Statistics (pp. 3118-3127). PMLR.

Using causal graphs to represent knowledge about shifts.
- Subbaswamy, A., Schulam, P., & Saria, S. (2019, April). Preventing failures due to dataset shift: Learning predictive models that transport. In The 22nd International Conference on Artificial Intelligence and Statistics (pp. 3118-3127). PMLR.
- Pearl, J., & Bareinboim, E. (2011, August). Transportability of causal and statistical relations: A formal approach. In Twenty-fifth AAAI conference on artificial intelligence.

As a very minor note: On line 16, reference [47] actually coins the term "dataset shift" for the "distribution shift" problem the authors refer to.

### UPDATES:
Thanks to the authors for their response. I think the practical discussion of the radius (i.e., its interpretation and general guidance on how to select it) should be included in the main paper. Even better would be to include or modify an experiment to demonstrate the (subjective) process by which a user would select a radius. Contingent on the inclusion of this practical discussion (as well as the expansion of the related work), I am raising my score 1 point.

**Time Spent Reviewing:**

4

---

> ### Author Response · Authors · 2021-08-09
> **Reply to Reviewer chLA**
>
> We thank the referee for the constructive feedback and the time spent in reviewing our paper.
>
> *Optimal choice of radius (size of ambiguity set).*
> Theorem 4.2 shows that the ambiguity set used in our paper displays a strong Pareto-optimality property, i.e., it leads to the least conservative predictor, uniformly across all estimator realizations, for which the out-of-sample disappointment probability is guaranteed to decay exponentially at rate $r$. Therefore, the radius $r$ has a direct operational interpretation that captures the risk tolerance of the decision maker---it is chosen subjectively.
> Since our statistical guarantees (Theorem 4.1 and Corollary 4.3) are tight only in the asymptotic regime, the reviewer is right in that the selection of the radius $r$ for finitely many data is a challenge and in practice is usually selected via cross validation. We hope that our provided operational interpretation of the hyperparameter $r$ can contribute toward that (still open) research question.
> We would like to add that our asymptotic consistency result (Theorem 4.4) suggests that for the out-of-sample disappointment to vanish in the limit, the radius $r_N$ should decay slower than $1/N$.
>
> *Reference Chen et al. (2018).*
> We thank the reviewer for this relevant reference. Indeed, the approach taken in this paper is related to ours, as it proposes a minimax approach for regression problems under covariate shift. The considered DRO framework optimizes over so-called moment-based ambiguity sets. Our approach is fundamentally different in two aspects: First, our framework is not specifically tailored to covariate shifts, these are rather one possible instance for our framework. Second, our choice of ambiguity set is statistically optimal in the sense of being Pareto-optimal as described in Theorem 4.2.
>
>
> *Discussion of related work regarding distribution shifts, causality and distributional robustness.*
> We are grateful to the reviewer for pointing out this interesting link and for the list of references. We will include a detailed discussion in our revised manuscript given the additional page.

---

### Decision · Program_Chairs · 2021-09-27

**Decision:**

Accept (Poster)

**Comment:**

Authors propose a DRO formulation that incorporates prior information in the form of moments of functionals, using minimum discriminating information. I agree with the reviewers that the new formulation is a meaningful contribution.

One remaining concern is that prior information studied in the experiments is somewhat contrived. The narrative of the paper will be substantially stronger if authors can demonstrate their method on a realistic example where the prior information presents naturally.